# TEST TIME ROBUSTIFICATION OF DEEP MODELS VIA ADAPTATION AND AUGMENTATION

## ABSTRACT

While deep neural networks can attain good accuracy on in-distribution test points, many applications require robustness even in the face of unexpected perturbations in the input, changes in the domain, or other sources of distribution shift. We study the problem of *test time robustification*, i.e., using the test input to improve model robustness. Recent prior works have proposed methods for test time adaptation, however, they each introduce additional assumptions, such as access to multiple test points, that prevent widespread adoption. In this work, we aim to study and devise methods that make no assumptions about the model training process and are broadly applicable at test time. We propose a simple approach that can be used in any test setting where the model is probabilistic and adaptable: when presented with a test example, perform different data augmentations on the data point, and then adapt (all of) the model parameters by minimizing the entropy of the model's average, or *marginal*, output distribution across the augmentations. Intuitively, this objective encourages the model to make the same prediction across different augmentations, thus enforcing the invariances encoded in these augmentations, while also maintaining confidence in its predictions. In our experiments, we evaluate two baseline ResNet models, two robust ResNet-50 models, and a robust vision transformer model, and we demonstrate that this approach achieves accuracy gains of 1-8% over standard model evaluation and also generally outperforms prior augmentation and adaptation strategies. For the setting in which only one test point is available, we achieve state-of-the-art results on the ImageNet-C, ImageNet-R, and, among ResNet-50 models, ImageNet-A distribution shift benchmarks.

## 1 INTRODUCTION

Deep neural network models have achieved excellent performance on many machine learning problems, such as image classification, but are often brittle and susceptible to issues stemming from *distribution shift*. For example, deep image classifiers may degrade precipitously in accuracy when encountering input perturbations, such as noise or changes in lighting (Hendrycks & Dietterich, 2019) or domain shifts which occur naturally in real world applications (Koh et al., 2021). Therefore, robustification of deep models against these test shifts is an important and active area of study.

Most prior works in this area have focused on techniques for training time robustification, including utilizing larger models and datasets (Orhan, 2019), various forms of adversarial training (Sagawa et al., 2020; Wong et al., 2020), and aggressive data augmentation (Yin et al., 2019; Hendrycks et al., 2020; Li et al., 2021; Hendrycks et al., 2021a). Employing these techniques requires modifying the training process, which may not be feasible if, e.g., it involves heavy computation or non-public data. Furthermore, these techniques do not rely on any information about the test points that the model must predict on, even though these test points may provide significant information for improving model robustness. Recently, several works have proposed methods for improving accuracy via *adaptation* after seeing the test data, typically by updating a subset of the model's weights (Sun et al., 2020), normalization statistics (Schneider et al., 2020), or both (Wang et al., 2021; Zhang et al., 2021). Though effective at handling test shifts, these methods sometimes still require specialized training procedures, and they typically rely on extracting distributional information via batches or even entire sets of test inputs, thus introducing additional assumptions.

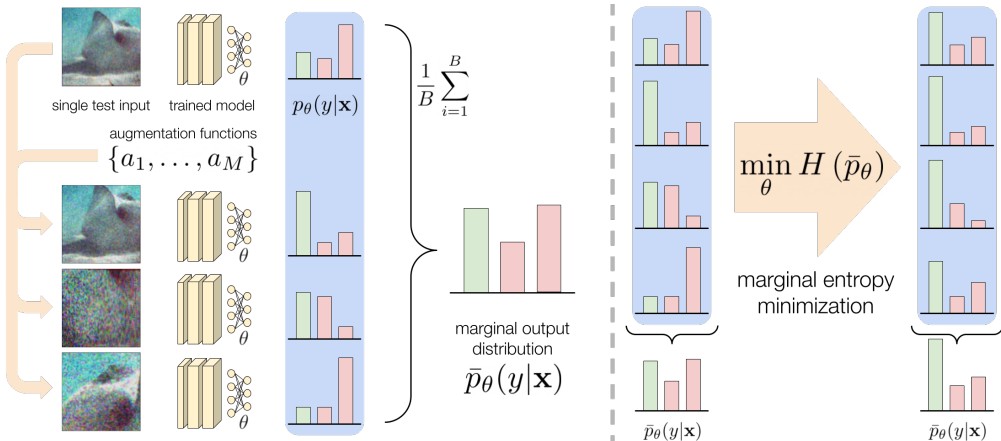

Figure 1: A schematic of our overall approach. Left: at test time, as detailed in Section 3, we have a trained model that outputs a probabilistic predictive distribution and has adaptable parameters $\theta$, a single test input $\mathbf{x}$, and a set of data augmentation functions $\{a_1, \ldots, a_M\}$. Note that we do not assume access to the model training procedure or multiple test inputs for adaptation. We perform different augmentations to $\mathbf{x}$ and pass these augmented inputs to the model in order to estimate the marginal output distribution averaged over augmentations. Right: rather than using this distribution to make the final prediction, we instead perform a gradient update on the model to minimize the entropy of this marginal distribution, thus encouraging the model predictions to be invariant across different augmentations while maintaining confident predictions. The final prediction is then made on the original data point, i.e., the predictive distribution in the top right of the schematic.

We are interested in studying and devising methods for improving model robustness that are "plug and play", i.e., they can be readily used with a wide variety of pretrained models and test settings. Such methods may simply constitute a different, more robust way to perform inference with preexisting models, under virtually the same assumptions as standard test time inference. In this work, we focus on methods for *test time robustness*, in which the specific test input may be leveraged in order to improve the model's prediction on that point. Though broad applicability is our primary goal, we also want methods that synergize with other robustification techniques, in order to achieve greater performance than using either set of techniques in isolation. To satisfy both of these desiderata, we devise a novel test time robustness method based on adaptation and augmentation. As illustrated in Figure 1, when presented with a test point, we propose to adapt the model by augmenting the test point in different ways and ensuring that the model makes the same predictions across these augmentations, thus respecting the invariances encoded in the data augmentations. We further encourage the model to make confident predictions, thus arriving at the proposed method: minimize the *marginal entropy* of the model's predictions across the augmented versions of the test point.

We refer to the proposed method as **m**arginal **e**ntropy **m**inimization with **e**nsembled augmentations (MEME), and this is the primary contribution of our work. MEME makes direct use of pretrained models without any assumptions about their particular training procedure or architecture, while requiring only a single test input for adaptation. In Section 4, we demonstrate empirically that MEME consistently improves the performance of ResNet (He et al., 2016) and vision transformer (Dosovitskiy et al., 2021) models on several challenging ImageNet distribution shift benchmarks, achieving several new state-of-the-art results for these models in the setting in which only one test point is available. In particular, MEME consistently outperforms non adaptive marginal distribution predictions (between 1-10% improvement) on corruption and rendition shifts – tested by the ImageNet-C (Hendrycks & Dietterich, 2019) and ImageNet-R (Hendrycks et al., 2021a) datasets, respectively – indicating that adaptation plays a crucial role in improving predictive accuracy. MEME encourages both invariance across augmentations and confident predictions, and an ablation study in Section 4 shows that both components are important for maximal performance gains. Also, MEME is, to the best of our knowledge, the first adaptation method to improve performance (by 1-4% over standard model evaluation) on ImageNet-A (Hendrycks et al., 2021b), demonstrating that MEME is more broadly applicable on a wide range of distribution shifts.

## 2 RELATED WORK

The general problem of distribution shift has been studied under a number of frameworks (Quiñonero Candela et al., 2009), including domain adaptation (Shimodaira, 2000; Csurka, 2017; Wilson & Cook, 2020), domain generalization (Blanchard et al., 2011; Muandet et al., 2013; Gulrajani & Lopez-Paz, 2021), and distributionally robust optimization (Ben-Tal et al., 2013; Hu et al., 2018; Sagawa et al., 2020), to name just a few. These frameworks typically leverage additional training or test assumptions in order to make the distribution shift problem more tractable. Largely separate from these frameworks, various empirical methods have also been proposed for dealing with shift, such as increasing the model and training dataset size or using heavy training augmentations (Orhan, 2019; Yin et al., 2019; Hendrycks et al., 2021a). The focus of this work is complementary to these efforts: the proposed MEME method is applicable to a wide range of pretrained models, including those trained via robustness methods, and can achieve further performance gains via test time adaptation.

Prior test time adaptation methods generally either make significant training or test time assumptions. Some methods update the model using batches or even entire datasets of test inputs, such as by computing batch normalization (BN) statistics on the test set (Li et al., 2017; Kaku et al., 2020; Nado et al., 2020; Schneider et al., 2020), or minimizing the (conditional) entropy of model predictions across a batch of test data (Wang et al., 2021). The latter approach is closely related to MEME. The differences are that MEME minimizes *marginal* entropy using single test points and data augmentation and adapts all of the model parameters rather than just those associated with normalization layers, thus not requiring multiple test points or specific model architectures. Other test time adaptation methods can be applied to single test points but require specific training procedures or models (Sun et al., 2020; Huang et al., 2020; Schneider et al., 2020). Test time training (TTT) (Sun et al., 2020) requires a specialized model with a rotation prediction head, as well as a different procedure for training this model. Schneider et al. (2020) show that BN adaptation can be effective even with only one test point, and we refer to this approach as "single point" BN adaptation. As we discuss in Section 3, MEME synergizes well with single point BN adaptation.

A number of works have noted that varying forms of strong data augmentation on the training set can improve the resulting model's robustness (Yin et al., 2019; Hendrycks et al., 2020; Li et al., 2021; Hendrycks et al., 2021a). Data augmentations are also sometimes used on the test data directly by averaging the model's outputs across augmented copies of the test point (Krizhevsky et al., 2012; Shorten & Khoshgoftaar, 2019), i.e., predicting according to the model's marginal output distribution. This technique, which we refer to as test time augmentation (TTA), has been shown to be useful both for improving model accuracy and calibration (Ashukha et al., 2020) as well as handling distribution shift (Molchanov et al., 2020). We take this idea one step further by explicitly adapting the model such that its marginal output distribution has low entropy. This extracts an additional learning signal for improving the model, and furthermore, the adapted model can then make its final prediction on the clean test point rather than the augmented copies. We empirically show in Section 4 that these differences lead to improved performance over this non adaptive TTA baseline.

## 3 ROBUSTNESS VIA ADAPTATION AND AUGMENTATION

Data augmentations are typically used to train the model to respect certain invariances – e.g., changes in lighting or viewpoint do not change the underlying class label – but, especially when faced with distribution shift, the model is not guaranteed to obey the same invariances at test time. In this section, we introduce MEME, a method for test time robustness that adapts the model such that it respects these invariances on the test input. We use "test time robustness" specifically to refer to techniques that operate directly on pretrained models and single test inputs – single point BN adaptation and TTA, as described in Section 2, are examples of prior test time robustness methods.

In the test time robustness setting, we are given a trained model $f_\theta : \mathcal{X} \to \mathcal{Y}$ with parameters $\theta \in \Theta$. We do not require any special training procedure and do not make any assumptions about the model, except that $\theta$ is adaptable and that $f_\theta$ produces a conditional output distribution $p_\theta(y|\mathbf{x})$ that is differentiable with respect to $\theta$.[1] All standard deep neural network models satisfy these assumptions. A single point $\mathbf{x} \in \mathcal{X}$ is presented to $f_\theta$, for which it must predict a label $\hat{y} \in \mathcal{Y}$ immediately. Note

---

[1] Single point BN adaptation also assumes that the model has batch normalization layers, and, as shown empirically in Section 4, this is an assumption that we do not require but can also benefit from.

---

**Algorithm 1** Test time robustness via MEME

---

**Require:** trained model $f_\theta$, test point $\mathbf{x}$, # augmentations $B$, learning rate $\eta$, update rule $G$

1: Sample $a_1, \ldots, a_B \overset{\text{i.i.d.}}{\sim} \mathcal{U}(\mathcal{A})$ and produce augmented points $\tilde{\mathbf{x}}_i = a_i(\mathbf{x})$ for $i \in \{1, \ldots, B\}$

2: Compute Monte Carlo estimate $\tilde{p} = \frac{1}{B} \sum_{i=1}^{B} p_\theta(y|\tilde{\mathbf{x}}_i) \approx \bar{p}_\theta(y|\mathbf{x})$ and $\tilde{\ell} = H(\tilde{p}) \approx \ell(\theta; \mathbf{x})$

3: Adapt model parameters via update rule $\theta' \leftarrow G(\theta, \eta, \tilde{\ell})$

4: Predict $\hat{y} \triangleq \arg\max_y p_{\theta'}(y|\mathbf{x})$

---

that this is precisely identical to the standard test time inference procedure for regular supervised learning models – in effect, we are simply modifying how inference is done, without any additional assumptions on the training process or on test time data availability. This makes test time robustness methods a simple "slot-in" replacement for the ubiquitous and standard test time inference process. We assume sampling access to a set of augmentation functions $\mathcal{A} \triangleq \{a_1, \ldots, a_M\}$ that can be applied to the test point $\mathbf{x}$. We use these augmentations and the self-supervised objective detailed below to adapt the model before it predicts on $\mathbf{x}$. When given a set of test inputs, the model adapts and predicts on each test point independently. We do not assume access to any ground truth labels.

### 3.1 MARGINAL ENTROPY MINIMIZATION WITH ENSEMBLED AUGMENTATIONS

Given a test point $\mathbf{x}$ and set of augmentation functions $\mathcal{A}$, we sample $B$ augmentations from $\mathcal{A}$ and apply them to $\mathbf{x}$ in order to produce a batch of augmented data $\tilde{\mathbf{x}}_1, \ldots, \tilde{\mathbf{x}}_B$. The model's average, or *marginal*, output distribution with respect to the augmented points is given by

$$\bar{p}_\theta(y|\mathbf{x}) \triangleq \mathbb{E}_{\mathcal{U}(\mathcal{A})}\left[p_\theta(y|a(\mathbf{x}))\right] \approx \frac{1}{B}\sum_{i=1}^{B} p_\theta(y|\tilde{\mathbf{x}}_i), \tag{1}$$

where the expectation is with respect to uniformly sampled augmentations $a \sim \mathcal{U}(\mathcal{A})$.

What properties do we desire from this marginal distribution? To answer this question, consider the role that data augmentation typically serves during training. For each training point $(\mathbf{x}^{\text{train}}, y^{\text{train}})$, the model $f_\theta$ is trained using multiple augmented forms of the input $\tilde{\mathbf{x}}_1^{\text{train}}, \ldots, \tilde{\mathbf{x}}_E^{\text{train}}$. $f$ is trained to obey the invariances between the augmentations and the label – no matter the augmentation on $\mathbf{x}^{\text{train}}$, $f$ should predict the same label $y^{\text{train}}$, and it should do so confidently. We seek to devise a similar learning signal during test time, when no ground truth labels are available. That is, after adapting:

(1) the model $f_\theta$ predictions should be invariant across augmented versions of the test point, and

(2) the model $f_\theta$ should be confident in its predictions, even for heavily augmented versions of the test point, due to the additional knowledge that all versions have the same underlying label.

Optimizing the model for more confident predictions can be justified from the assumption that the true underlying decision boundaries between classes lie in low density regions of the data space (Grandvalet & Bengio, 2005). With these two goals in mind, we propose to adapt the model using the entropy of its marginal output distribution over augmentations (Equation 1), i.e.,

$$\ell(\theta; \mathbf{x}) \triangleq H\left(\bar{p}_\theta(\cdot|\mathbf{x})\right) = -\sum_{y \in \mathcal{Y}} \bar{p}_\theta(y|\mathbf{x}) \log \bar{p}_\theta(y|\mathbf{x}). \tag{2}$$

Optimizing this objective encourages both confidence and invariance to augmentations, since the entropy of $\bar{p}_\theta(\cdot|\mathbf{x})$ is minimized when the model outputs the same (confident) prediction regardless of the augmentation. Given that $\theta$ is adaptable and $p_\theta(y|\mathbf{x})$ is differentiable with respect to $\theta$, we can directly use gradient based optimization to adapt $\theta$ according to this objective. We use only one gradient step per test point, because empirically we found this to be sufficient for improved performance while being more computationally efficient. After this step, we can use the adapted model, which we denote $f_{\theta'}$, to predict on the original test input $\mathbf{x}$.

Algorithm 1 presents the overall method MEME for test time adaptation. Though prior test time adaptation methods must carefully choose which parameters to adapt in order to avoid degenerate solutions (Wang et al., 2021), our adaptation procedure simply adapts all of the model's parameters $\theta$ (line 3). Note that, as discussed above, the model $f_\theta$ adapts using augmented data but makes its final prediction on the original point (line 4), which may be easier to predict on.

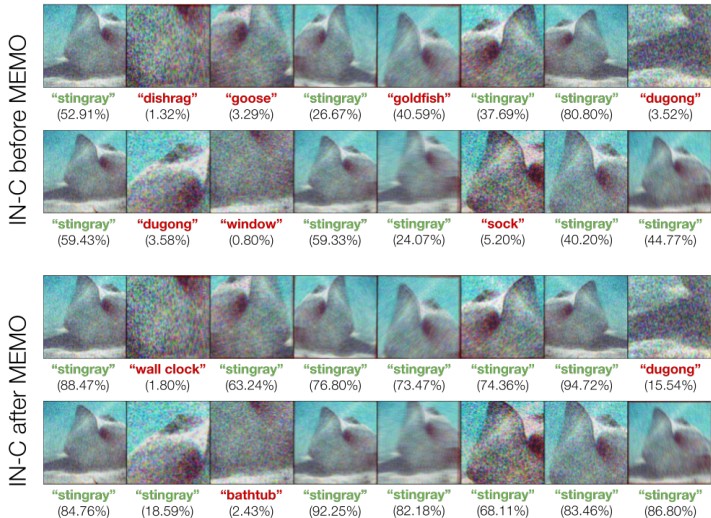

Figure 2: We visualize augmentations of a randomly chosen data point from the "Gaussian Noise level 3" ImageNet-C test set. Even for a robust model trained with heavy data augmentations (Hendrycks et al., 2021a), both its predictive accuracy and confidence drop sharply when encountering test distribution shift. As shown in the bottom two rows, these drops can be remedied via MEME.

## 3.2 COMPOSING MEME WITH PRIOR METHODS

An additional benefit of MEME is that it synergizes with other approaches for handling distribution shift. In particular, MEME can be composed with prior methods for training robust models and adapting model statistics, thus leveraging the performance improvements of each technique.

**Pretrained robust models.** Since MEME makes no assumptions about, or modifications to, the model training procedure, performing adaptation on top of pretrained robust models, such as those trained with heavy data augmentations, is as simple as using any other pretrained model. Crucially, we find that, in practice, the set of augmentations that we use at test time $\mathcal{A}$ does not have to match the augmentations that were used to train the model. This is important as we require a few properties from the test time augmentations: that they can be easily sampled and are applied directly to the model input $\mathbf{x}$. These properties do not hold for, e.g., data augmentation techniques based on image translation models, such as DeepAugment (Hendrycks et al., 2021a), or feature mixing, such as moment exchange (Li et al., 2021). However, we can still use models trained with these data augmentation techniques as our starting point for adaptation, thus allowing us to improve upon their state-of-the-art results. As noted above, using pretrained models is not as easily accomplished for adaptation methods which require complicated or specialized training procedures and model architectures, such as TTT (Sun et al., 2020) or ARM (Zhang et al., 2021). In our experiments, we use AugMix as our set of augmentations (Hendrycks et al., 2020), as it satisfies the above properties and still yields significant diversity when applied, as depicted in Figure 2.

**Adapting BN statistics.** Schneider et al. (2020) showed that, even when presented with just a single test point, partially adapting the estimated mean and variance of the activations in each batch normalization (BN) layer of the model can still be effective in some cases for handling distribution shift. In this setting, to prevent overfitting to the test point, the channelwise mean and variance $[\boldsymbol{\mu}_{\text{test}}, \boldsymbol{\sigma}_{\text{test}}^2]$ estimated from this point are mixed with the mean and variance $[\boldsymbol{\mu}_{\text{train}}, \boldsymbol{\sigma}_{\text{train}}^2]$ computed during training according to a prior strength $N$, i.e.,

$$\boldsymbol{\mu} \triangleq \frac{N}{N+1}\boldsymbol{\mu}_{\text{train}} + \frac{1}{N+1}\boldsymbol{\mu}_{\text{test}}, \boldsymbol{\sigma}^2 \triangleq \frac{N}{N+1}\boldsymbol{\sigma}_{\text{train}}^2 + \frac{1}{N+1}\boldsymbol{\sigma}_{\text{test}}^2.$$

This technique is also straightforward to combine with MEME: we simply use the adapted BN statistics whenever computing the model's output distribution. That is, we adapt the BN statistics alongside all of the model parameters for MEME. Following the suggestion in Schneider et al. (2020), we set $N = 16$ for all of our experiments in the next section.

## 4 EXPERIMENTS

Our experiments aim to answer the following questions:

(1) How does MEME compare to prior methods for test time adaptation, which make additional training and test assumptions, and test time robustness?

(2) Can MEME be successfully combined with a wide range of pretrained models?

(3) Which aspects of MEME are the most important for strong performance?

We conduct experiments on two distribution shift benchmarks for CIFAR-10 (Krizhevsky, 2009) and three distribution shift benchmarks for ImageNet (Russakovsky et al., 2015). Specifically, for CIFAR-10, we evaluate on the CIFAR-10-C (Hendrycks & Dietterich, 2019) and CIFAR-10.1 (Recht et al., 2018) test sets, and for ImageNet, we evaluate on the ImageNet-C (Hendrycks & Dietterich, 2019), ImageNet-R (Hendrycks et al., 2021a), and ImageNet-A (Hendrycks et al., 2021b) test sets.

To answer question (1), we compare to test time training (TTT) (Sun et al., 2020) in the CIFAR-10 experiments, for which we train ResNet-26 models following their protocol and specialized architecture. We do not compare to TTT for the ImageNet experiments due to the computational demands of training state-of-the-art models and because Sun et al. (2020) do not report competitive ImageNet results. For the ImageNet experiments, we compare to Tent (Wang et al., 2021) and BN adaptation, which can be used with pretrained models but require multiple test inputs (or even the entire test set) for adaptation. We provide BN adaptation with $256$ test inputs at a time and, following Schneider et al. (2020), set the prior strength $N = 256$ accordingly.

For Tent, we use test batch sizes of $64$ and, for ResNet-50 models, test both "online" adaptation – where the model adapts continually through the entire evaluation – and "episodic" adaptation – where the model is reset after each test batch. (Wang et al., 2021). Note that the evaluation protocols are different for these two methods: whereas MEME is tasked with predicting on each test point immediately after adaptation, BN adaptation predicts on a batch of 256 test points after computing BN statistics on the batch, and Tent predicts on a batch of 64 inputs after adaptation but also, in the online setting, continually adapts throughout evaluation. In all experiments, we further compare to single point BN adaptation (Schneider et al., 2020) and the TTA baseline that simply predicts according to the model's marginal output distribution over augmentations $\bar{p}_\theta(y|\mathbf{x})$ (Equation 1) (Krizhevsky et al., 2012; Ashukha et al., 2020). Full details on our experimental protocol are provided in Appendix A.

To answer question (2), we apply MEME on top of multiple pretrained models with different architectures, trained via several different procedures. For CIFAR-10, we train our own ResNet-26 (He et al., 2016) models. For ImageNet, we use the best performing ResNet-50 robust models from prior work, which includes those trained with DeepAugment and AugMix augmentations (Hendrycks et al., 2021a) as well as those trained with moment exchange and CutMix (Li et al., 2021). To evaluate the generality of prior test time robustness methods and MEME, we also evaluate the small robust vision transformer (RVT*-small), which provides superior performance on all three ImageNet distribution shift benchmarks compared to the robust ResNet-50 models (Mao et al., 2021).

Finally, to answer (3), we conduct ablative studies in subsection 4.2: first to determine the relative importance of maximizing confidence (via entropy minimization) versus enforcing invariant predictions across augmented copies of each test point, and second to determine the importance of the particular augmentation functions used. The comparison to the non adaptive TTA baseline also helps determine whether simply augmenting the test point is sufficient or if adaptation is additionally helpful.

### 4.1 MAIN RESULTS

We summarize results for CIFAR-10, CIFAR-10.1, and CIFAR-10-C in Table 1, with full CIFAR-10-C results in Appendix C. We use indentations to indicate composition, e.g., TTT is performed at test time on top of their specialized joint training procedure. Across all corruption types in CIFAR-10-C, MEME consistently improves test error compared to the baselines, non adaptive TTA, and TTT. MEME also provides a larger performance gain on CIFAR-10.1 compared to TTT. We find that the non adaptive TTA baseline is competitive for these relatively simple test sets, though it is worse than MEME for CIFAR-10-C. Of these three test sets, CIFAR-10-C is the only benchmark that explicitly introduces distribution shift, which suggests that adaptation is useful when the test shifts are

Table 1: Results for the original CIFAR-10 test set, CIFAR-10.1, and CIFAR-10-C. MEME outperforms TTT despite not making any training assumptions. *Results from Sun et al. (2020).

|  | CIFAR-10 Error (%) | CIFAR-10.1 Error (%) | CIFAR-10-C Average Error (%) |
|---|---|---|---|
| ResNet-26 (He et al., 2016) | 9.2 | 18.4 | 22.5 |
| + TTA | **7.3** (−1.9) | 14.8 (−3.6) | 19.9 (−2.6) |
| + MEME (ours) | **7.3** (−1.9) | **14.7** (−3.7) | **19.6** (−2.9) |
| + Joint training* (Sun et al., 2020) | 8.1 | 16.7 | 22.8 |
| + TTT* (Sun et al., 2020) | 7.9 (−0.2) | 15.9 (−0.8) | 21.5 (−1.3) |

Table 2: Test results for ImageNet-C, ImageNet-R, and ImageNet-A. MEME achieves new state-of-the-art performance on each benchmark for ResNet-50 models for the single test point setting. For RVT*-small, MEME substantially improves performance across all benchmarks and reaches a new state of the art for ImageNet-C and ImageNet-R.

|  | ImageNet-C mCE ↓ | ImageNet-R Error (%) | ImageNet-A Error (%) |
|---|---|---|---|
| Baseline ResNet-50 (He et al., 2016) | 76.7 | 63.9 | 100.0 |
| + TTA | 77.9 (+1.2) | 61.3 (−2.6) | 98.4 (−1.6) |
| + Single point BN | 71.4 (−5.3) | 61.1 (−2.8) | 99.4 (−0.6) |
| + MEME (ours) | 69.9 (−6.8) | 58.8 (−5.1) | 99.1 (−0.9) |
| + BN ($N = 256, n = 256$) | 61.6 (−15.1) | 59.7 (−4.2) | 99.8 (−0.2) |
| + Tent (online) (Wang et al., 2021) | 54.4 (−22.3) | 57.7 (−6.2) | 99.8 (−0.2) |
| + Tent (episodic) | 64.7 (−12.0) | 61.0 (−2.9) | 99.7 (−0.3) |
| + DeepAugment+AugMix (Hendrycks et al., 2021a) | 53.6 | 53.2 | 96.1 |
| + TTA | 55.2 (+1.6) | 51.0 (−2.2) | 93.5 (−2.6) |
| + Single point BN | 51.3 (−2.3) | 51.2 (−2.0) | 95.4 (−0.7) |
| + MEME (ours) | **49.8** (−3.8) | **49.2** (−4.0) | 94.8 (−1.3) |
| + BN ($N = 256, n = 256$) | 45.4 (−8.2) | 48.8 (−4.4) | 96.8 (+0.7) |
| + Tent (online) | **43.5** (−10.1) | **46.9** (−6.3) | 96.7 (+0.6) |
| + Tent (episodic) | 47.1 (−6.5) | 50.1 (−3.1) | 96.6 (+0.5) |
| + MoEx+CutMix (Li et al., 2021) | 74.8 | 64.5 | 91.9 |
| + TTA | 75.7 (+0.9) | 62.7 (−1.8) | 89.5 (−2.4) |
| + Single point BN | 71.0 (−3.8) | 62.6 (−1.9) | 91.1 (−0.8) |
| + MEME (ours) | 69.1 (−5.7) | 59.4 (−3.3) | **89.0** (−2.9) |
| + BN ($N = 256, n = 256$) | 60.9 (−13.9) | 61.6 (−2.9) | 93.9 (+2.0) |
| + Tent (online) | 54.0 (−20.8) | 58.7 (−5.8) | 94.4 (+2.5) |
| + Tent (episodic) | 66.2 (−8.6) | 63.9 (−0.6) | 94.7 (+2.8) |
| RVT*-small (Mao et al., 2021) | 49.4 | 52.3 | 73.9 |
| + TTA | 53.0 (+3.6) | 49.0 (−3.3) | **68.9** (−5.0) |
| + Single point BN | 48.0 (−1.4) | 51.1 (−1.2) | 74.4 (+0.5) |
| + MEME (ours) | **40.6** (−8.8) | **43.8** (−8.5) | 69.8 (−4.1) |
| + BN ($N = 256, n = 256$) | 44.3 (−5.1) | 51.0 (−1.3) | 78.3 (+4.4) |
| + Tent (online) | 46.8 (−2.6) | 50.7 (−1.6) | 82.1 (+8.2) |
| + Tent (adapt all) | 44.7 (−4.7) | 74.1 (+21.8) | 81.1 (+7.2) |

more prominent. Both TTA and MEME are also effective at improving performance for the original CIFAR-10 test set where there is no distribution shift, providing further support for the widespread use of augmentations in standard evaluation protocols (Krizhevsky et al., 2012; Ashukha et al., 2020).

We summarize results for ImageNet-C, ImageNet-R, and ImageNet-A in Table 2, with complete ImageNet-C results in Appendix C. We again use indentations to indicate composition, e.g., the best results on ImageNet-C for our setting are attained through a combination of starting from a model trained with DeepAugment and AugMix (Hendrycks et al., 2021a) and using MEME on

Table 3: Ablating the adaptation objective to test pairwise cross entropy and conditional entropy (CE) based adaptation. MEME generally performs the best, indicating that both encouraging invariance across augmentations and confidence are helpful in adapting the model.

|  | CIFAR-10 Error (%) | CIFAR-10.1 Error (%) | CIFAR-10-C Average Error (%) |
|---|---|---|---|
| ResNet-26 (He et al., 2016) | 9.2 | 18.4 | 22.5 |
| + MEME (ours) | **7.3** (−1.9) | **14.7** (−3.7) | **19.6** (−2.9) |
| − $\ell$ (Equation 2) + $\ell_{\mathrm{PCE}}$ | 7.6 (−1.6) | 15.3 (−3.1) | 20.0 (−2.5) |
| − $\ell$ (Equation 2) + $\ell_{\mathrm{CE}}$ | 7.6 (−1.6) | **14.7** (−3.7) | 20.0 (−2.5) |

|  | ImageNet-C mCE ↓ | ImageNet-R Error (%) | ImageNet-A Error (%) |
|---|---|---|---|
| RVT*-small (Mao et al., 2021) | 49.4 | 52.3 | 73.9 |
| + MEME (ours) | **40.6** (−8.8) | **43.8** (−8.5) | 69.8 (−4.1) |
| − $\ell$ (Equation 2) + $\ell_{\mathrm{CE}}$ | 41.2 (−8.2) | 44.2 (−8.1) | **69.7** (−4.2) |

top. For both ImageNet-C and ImageNet-R, and for both the ResNet-50 and RVT*-small models, combining MEME with robust training techniques leads to new state of the art performance among methods that observe only one test point at a time. We highlight in gray the methods that require multiple test points for adaptation, and we list in bold the best results from these methods which outperform the test time robustness methods. As Table 2 and prior work both show (Schneider et al., 2020; Wang et al., 2021), accessing multiple test points can be powerful for benchmarks such as ImageNet-C and ImageNet-R, in which inferred statistics from the test input distribution may aid in prediction. In the case of Tent, which adapts online, the model has adapted using the entire test set by the end of evaluation. However, these methods do not help, and oftentimes even hurt, for ImageNet-A. Furthermore, we find that these methods are less effective with the RVT*-small model, which may indicate their sensitivity to model architecture choices. Therefore, for this model, we also test a modification of Tent which adapts all parameters, and we find that this version of Tent works better for ImageNet-C but is significantly worse for ImageNet-R.

MEME results in substantial improvement for ImageNet-A and is competitive with TTA on this problem. No prior test time adaptation methods have reported improvements on ImageNet-A, and some have reported explicit negative results (Schneider et al., 2020). As discussed, it is reasonable for adaptation methods that rely on multiple test points to achieve greater success on other benchmarks such as ImageNet-C, in which a batch of inputs provides significant information about the specific corruption that must be dealt with. In contrast, ImageNet-A does not have such obvious characteristics associated with the input distribution, as it is simply a collection of images that are difficult to classify. As MEME instead extracts a learning signal from single test points, it is, to the best of our knowledge, the first test time adaptation method to report successful results on this testbed. When used on top of a model trained with moment exchange and CutMix (Li et al., 2021), MEME achieves state-of-the-art performance among ResNet-50 models and single test point methods. TTA generally offers larger performance gains on ImageNet-A and also results in the highest overall accuracy when combined with the RVT*-small model; however, TTA performs worse than MEME on ImageNet-R and consistently *decreases* accuracy on ImageNet-C compared to standard evaluation. We view the consistency with which MEME outperforms the best prior methods, which change across different test sets, to be a major advantage of the proposed method.

## 4.2 ABLATIVE STUDY

MEME increases model robustness at test time via adaptation and augmentation. In this section, we ablate the adaptation procedure, and in Appendix B we ablate the use and choice of augmentations.

From the results above, we conclude that adaptation generally provides additional benefits beyond simply using TTA to predict via the marginal output distribution $\bar{p}_\theta(y|\mathbf{x})$. However, we can disentangle two distinct self-supervised learning signals that may be effective for adaptation: encouraging invariant predictions across different augmentations of the test point, and encouraging confidence via entropy minimization. The marginal entropy objective in Equation 2 encapsulates both of these

learning signals, but it cannot easily be decomposed into these pieces. Thus, we instead use two ablative adaptation methods that each only make use of one of these learning signals.

First, we consider optimizing the pairwise cross entropy between each pair of augmented points, i.e.,

$$\ell_{\text{PCE}}(\theta; \mathbf{x}) \triangleq \frac{1}{B \times (B-1)} \sum_{i=1}^{B} \sum_{j \neq i} \sum_{y \in \mathcal{Y}} p_\theta(y|\tilde{\mathbf{x}}_i) \log p_\theta(y|\tilde{\mathbf{x}}_j) \,,$$

Where $\tilde{\mathbf{x}}_i$ again refers to the $i$-th sampled augmentation applied to $\mathbf{x}$. Intuitively, this loss function encourages the model to adapt such that it produces the same predictive distribution for all augmentations of the test point, but it does not encourage the model to produce confident predictions. Conversely, as an objective that encourages confidence but not invariance, we also consider optimizing conditional entropy on the batch of augmented points, i.e.,

$$\ell_{\text{CE}}(\theta; \mathbf{x}) \triangleq \frac{1}{B} \sum_{i=1}^{B} H(p_\theta(\cdot|\tilde{\mathbf{x}}_i)) \,.$$

This ablation is effectively a version of the episodic variant of Tent (Wang et al., 2021) that produces augmented copies of a single test point rather than assuming access to a test batch. We first evaluate these ablations on the CIFAR-10 test sets. We use the same adaptation procedure outlined in Algorithm 1, with $\ell$ replaced with the above objectives, and we keep the same hyperparameter values.

The results are presented in Table 3. We see that MEME, i.e., marginal entropy minimization, generally performs better than adaptation with either of the alternative objectives. This supports the hypothesis that both invariance across augmentations and confidence are important learning signals for self-supervised adaptation. When faced with CIFAR-10.1, we see poor performance from the pairwise cross entropy based adaptation method. On the original CIFAR-10 test set and CIFAR-10-C, the ablations perform nearly identically and uniformly worse than MEME. To further test the $\ell_{\text{CE}}$ ablation, we also evaluate it on the ImageNet test sets for the RVT*-small model. We find that, similarly, minimizing conditional entropy generally improves performance compared to the baseline evaluation. MEME is more performant for ImageNet-C and ImageNet-R, again indicating the benefits of encouraging invariance to augmentations. Adaptation via $\ell_{\text{CE}}$ performs slightly better for ImageNet-A, though for this problem, TTA is still the best method.

## 5 DISCUSSION

We presented MEME, a method for test time robustification again distribution shift via adaptation and augmentation. MEME does not require access or changes to the model training procedure and is thus broadly applicable for a wide range of pretrained models. Furthermore, MEME adapts at test time using single test inputs, thus it does not assume access to multiple test points as in several recent methods for test time adaptation (Schneider et al., 2020; Wang et al., 2021; Zhang et al., 2021). On a range of distribution shift benchmarks for CIFAR-10 and ImageNet classification, and for both ResNet and vision transformer models, MEME consistently improves performance at test time and achieves several new state-of-the-art results for these models in the single test point setting.

Inference via MEME is more computationally expensive than standard model inference, primarily because adaptation is performed per test point and thus inference cannot be batched. When deployed in the real world, it is natural to expect that test points will arrive one at a time and batched inference will not be possible. However, MEME is also more computationally expensive due to its augmentation and adaptation procedure. One interesting direction for future work is to develop techniques for selectively determining when to adapt the model in order to achieve more efficient inference. For example, with well calibrated models (Guo et al., 2017), we may run simple "feedforward" inference when the prediction confidence is over a certain threshold, thus achieving better efficiency. Additionally, it would be interesting to explore MEME in the test setting where the model is allowed to continually adapt as more test data is observed. In our preliminary experiments in this setting, MEME tended to lead to degenerate solutions, e.g., the model predicting a constant label with maximal confidence, and this may potentially be rectified by carefully choosing which parameters to adapt (Wang et al., 2021) or regularizing the model such that it does not change too drastically from the pretrained model.

## 6 ETHICS STATEMENT

Distribution shift, in general, lies at the heart of many ethical concerns in machine learning. When machine learning models and research do not adhere to ethical principles such as "contribute to society and to human well-being", "avoid harm", and "be fair and take action to avoid discrimination", it can oftentimes be attributed at least partially to issues stemming from distribution shift. As such, research into ways to combat and mitigate shift contribute to furthering the ethical goals and principles laid out by this conference and the broader community. Related to this work, we may imagine that models that are more robust or adaptable to shift may produce more fair results when considering underrepresented subpopulations and could be more trustworthy in safety critical applications. However, as mentioned in Section 5, the proposed method is also more computationally intensive, and care must be taken in general to not place the most powerful machine learning tools exclusively in the hands of the most privileged and resource rich individuals and organizations. Beyond this, we do not see any immediate ethical concerns regarding the content, presentation, or methodology of this work.

## 7 REPRODUCIBILITY STATEMENT

The proposed method is relatively simple and, to the best of our ability, explained fully in Section 3 and Algorithm 1. This explanation is bolstered by a complete description of all hyperparameters in Appendix A as well as the example code provided in the supplementary materials. As briefly explained in the README in the example code, all datasets were downloaded from publicly accessible links and preprocessed only following the standard protocols. Upon publication, we will include a link to the full (non anonymous) code release containing full instructions for setting up the code, downloading the datasets, and reproducing the results.

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

## A    EXPERIMENTAL PROTOCOL

We select hyperparameters using the four disjoint validation corruptions provided with CIFAR-10-C and ImageNet-C (Hendrycks & Dietterich, 2019). As the other benchmarks are only test sets and do not provide validation sets, we use the same hyperparameters found using the corruption validation sets and do not perform any additional tuning. For the ResNet models that we evaluate, we use stochastic gradients as the update rule $G$; for ResNet-26 models, we set the number of augmentations $B = 32$ and the learning rate $\eta = 0.005$; and for ResNet-50 models, we set $B = 64$ and $\eta = 0.00025$. For the robust vision transformer, we use AdamW (Loshchilov & Hutter, 2019) as the update rule $G$, with learning rate $\eta = 0.00001$ and weight decay $0.01$, and $B = 64$.

In the CIFAR evaluation, we compare to TTT, which, as noted, can also be applied to single test inputs but requires a specialized training procedure (Sun et al., 2020). Thus, the ResNet-26 model we use for our method closely follows the modifications that Sun et al. (2020) propose, in order to provide a fair point of comparison. In particular, Sun et al. (2020) elect to use group normalization (Wu & He, 2018) rather than BN, thus single point BN adaptation is not applicable for this model architecture. As noted before, TTT also requires the joint training of a separate rotation prediction head, thus further changing the model architecture, while MEME directly adapts the standard pretrained model.

The TTA results are obtained using the same AugMix augmentations as for MEME. The single point BN adaptation results use $N = 16$, as suggested by Schneider et al. (2020). As noted, the BN adaptation results (using multiple test points) are obtained using $N = 256$ as the prior strength and batches of 256 test inputs for adaptation. For Tent, we use the hyperparameters suggested in Wang et al. (2021): we use stochastic gradients with learning rate $0.00025$ and momentum $0.9$, the adaptation is performed with test batches of $64$ inputs, and the method is run online, i.e., prediction and adaptation occur simultaneously and the model is allowed to continuously adapt through the entire test epoch. Since Wang et al. (2021) did not experiment with transformer models, we also attempted to run Tent with Adam (Kingma & Ba, 2015) and AdamW (Loshchilov & Hutter, 2019) and various hyperparameters for the RVT*-small model; however, we found that this generally resulted in worse performance than using stochastic gradient updates with the aforementioned hyperparameters.

We obtain the baseline ResNet-50 parameters directly from the `torchvision` library. The parameters for the ResNet-50 trained with DeepAugment and AugMix are obtained from `https://drive.google.com/file/d/1QKmc_p6-qDkh51WvsaS9HKFv8bX5jLnP`. The parameters for the ResNet-50 trained with moment exchange and CutMix are obtained from `https://drive.google.com/file/d/1cCvhQKV93pY-jj8f5jITywkB9EabiQDA`. The parameters for the small robust vision transformer (RVT*-small) model are obtained from `https://drive.google.com/file/d/1g40huqDVthjS2H5sQV3ppcfcWEzn9ekv`.

## B    ADDITIONAL EXPERIMENTS

In this section, we analyze the importance of using augmentations during adaptation, study the trade-offs between efficiency and accuracy for MEME, and present results with ResNext101 models (Xie et al., 2017; Mahajan et al., 2018; Orhan, 2019) on ImageNet-A.

### B.1    ANALYSIS ON AUGMENTATIONS

One may first wonder: are augmentations needed in the first place? In the test time robustness setting when only one test point is available, how would simple entropy minimization fare? We answer this question in Table 4 by evaluating the episodic variant of Tent (i.e., with model resetting after each batch) with a test batch size of 1. This approach is also analogous to a variant of MEME that does not use augmentations, since for one test point and no augmented copies, conditional and marginal entropy are the same. Similar to MEME, we also incorporate single point BN adaptation with $N = 16$, in place of the standard BN adaptation that Tent typically employs using batches of test inputs. The results in Table 4 indicate that entropy minimization on a single test point generally provides no additional performance gains beyond just single point BN adaptation. This empirically shows that using augmentations is important for achieving the reported results.

We also wish to understand the importance of the choice of augmentation functions $\mathcal{A}$. As mentioned, we used AugMix (Hendrycks et al., 2020) in the previous experiments as it best fit our criteria:

Table 4: Evaluating the episodic version of Tent with a batch size of 1, which corresponds to a simple entropy minimization approach for the test time robustness setting. This approach also uses single point BN adaptation, and entropy minimization does not provide much, if any, additional gains.

| | ImageNet-C mCE ↓ | ImageNet-R Error (%) | ImageNet-A Error (%) |
|---|---|---|---|
| Baseline ResNet-50 (He et al., 2016) | 76.7 | 63.9 | 100.0 |
| + Single point BN | 71.4 (−5.3) | 61.1 (−2.8) | 99.4 (−0.6) |
| + MEME (ours) | 69.9 (−6.8) | 58.8 (−5.1) | 99.1 (−0.9) |
| + Tent (episodic, batch size 1) (Wang et al., 2021) | 71.4 (−5.3) | 61.2 (−2.7) | 99.4 (−0.6) |
| + DeepAugment+AugMix (Hendrycks et al., 2021a) | 53.6 | 53.2 | 96.1 |
| + Single point BN | 51.3 (−2.3) | 51.2 (−2.0) | 95.4 (−0.7) |
| + MEME (ours) | **49.8** (−**3.8**) | **49.2** (−**4.0**) | 94.8 (−1.3) |
| + Tent (episodic, batch size 1) (Wang et al., 2021) | 51.2 (−2.4) | 50.9 (−2.3) | 95.4 (−0.7) |
| + MoEx+CutMix (Li et al., 2021) | 74.8 | 64.5 | 91.9 |
| + Single point BN | 71.0 (−3.8) | 62.6 (−1.9) | 91.1 (−0.8) |
| + MEME (ours) | 69.1 (−5.7) | 59.4 (−3.3) | **89.0** (−**2.9**) |
| + Tent (episodic, batch size 1) (Wang et al., 2021) | 70.9 (−3.9) | 62.6 (−1.9) | 91.1 (−0.8) |
| RVT*-small (Mao et al., 2021) | 49.4 | 52.3 | 73.9 |
| + Single point BN | 48.0 (−1.4) | 51.1 (−1.2) | 74.4 (+0.5) |
| + MEME (ours) | **40.6** (−**8.8**) | **43.8** (−**8.5**) | **69.8** (−**4.1**) |
| + Tent (episodic, batch size 1) (Wang et al., 2021) | 48.1 (−1.3) | 50.7 (−1.6) | 74.4 (+0.5) |

Table 5: Ablating the augmentation functions to test standard augmentations (random resized cropping and horizontal flips). When changing the augmentations used, the post-adaptation performance generally does not change much, though it suffers the most on CIFAR-10-C.

| | CIFAR-10 Error (%) | CIFAR-10.1 Error (%) | CIFAR-10-C Average Error (%) |
|---|---|---|---|
| ResNet-26 (He et al., 2016) | 9.2 | 18.4 | 22.5 |
| + MEME (ours) | 7.3 (−1.9) | 14.7 (−3.7) | **19.6** (−**2.9**) |
| − AugMix (Hendrycks et al., 2020) + standard augs | **7.2** (−**2.0**) | **14.6** (−**3.8**) | 20.2 (−2.3) |

AugMix requires only the input **x**, and randomly sampled augmentations lead to diverse augmented data points. A simple alternative is to instead use the "standard" set of augmentations commonly used in ImageNet training, i.e., random resized cropping and random horizontal flipping. We evaluate this ablation of using MEME with standard augmentations also on the CIFAR-10 test sets, again with the same hyperparameter values. From the results in Table 5, we can see that MEME is still effective with simpler augmentation functions. This is true particularly for the cases where there is no test shift, as in the original CIFAR-10 test set, or subtle shifts as in CIFAR-10.1; however, for the more severe and systematic CIFAR-10-C shifts, using heavier AugMix data augmentations leads to greater performance gains over the standard augmentations. Furthermore, this ablation was conducted using the ResNet-26 model, which was trained with standard augmentations – for robust models such as those in Table 2, AugMix may offer greater advantages at test time since these models were exposed to heavy augmentations during training.

## B.2    ANALYZING THE TRADEOFF BETWEEN EFFICIENCY AND ACCURACY

In Figure 3, we analyze the % test error of MEME adaptation on ImageNet-R as a function of the efficiency of adaptation, measured in seconds per evaluation. We achieve various tradeoffs by varying the number of augmented copies $B = \{1, 2, 4, 8, 16, 32, 64, 128\}$. We note that small values of $B$ such as 4 and 8 can already provide significant performance gains, indicating that a practical tradeoff between efficiency and accuracy is possible. For large $B$, the wall clock time is dominated by computing the augmentations – in our implementation, we do not compute augmentations in

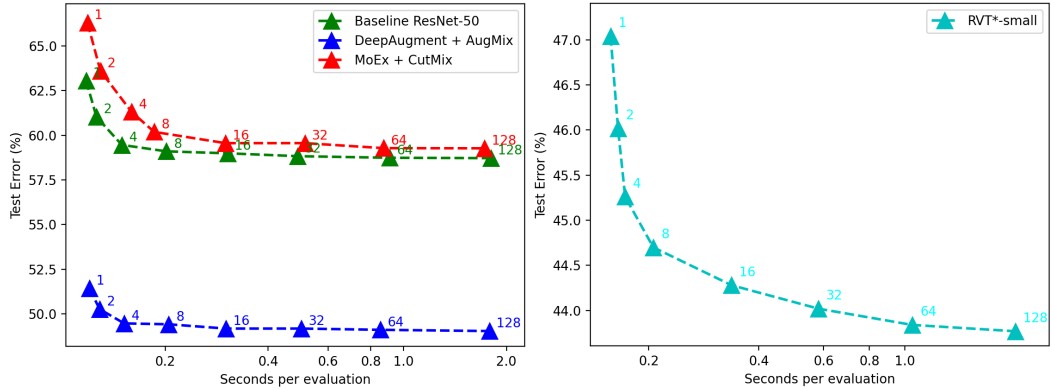

Figure 3: Plotting MEME efficiency as seconds per evaluation (x axis) and % test error on ImageNet-R (y axis) for the ResNet-50 models (left) and RVT*-small (right) while varying $B = \{1, 2, 4, 8, 16, 32, 64, 128\}$. Note the log scale on the x axis.

Table 6: ImageNet-A results for the ResNext-101 models.

|  | ImageNet-A Error (%) |
| --- | --- |
| Baseline ResNext-101 (32x8d) (Xie et al., 2017) | 90.0 |
|    + TTA | 83.2 (−6.8) |
|    + Single point BN | 88.8 (−1.2) |
|    + MEME (ours) | 84.3 (−5.7) |
| + WSL on billions of images (Mahajan et al., 2018) | 54.9 |
|    + TTA | 49.1 (−5.8) |
|    + MEME (ours) | **43.2** (−**11.7**) |

parallel, though in principle this is possible for AugMix and should improve efficiency overall. These experiments used four Intel Xeon Skylake 6130 CPUs and one NVIDIA TITAN RTX GPU.

## B.3 EVALUATING RESNEXT101 MODELS ON IMAGENET-A

ResNext-101 models (Xie et al., 2017) have been found to achieve higher accuracies on ImageNet-A (Hendrycks et al., 2021b), particularly when trained with massive scale weakly supervised pretraining (Mahajan et al., 2018; Hendrycks et al., 2021a). In this section, we evaluate whether MEME can successfully adapt these models and further improve performance on this challenging test set. We use the same hyperparameters as for the robust vision transformer with no additional tuning: AdamW (Loshchilov & Hutter, 2019) as the update rule $G$, learning rate $\eta = 0.00001$, weight decay 0.01, and $B = 64$. We obtain the baseline ResNext-101 (32x8d) parameters, pretrained on ImageNet, directly from the `torchvision` library. We also evaluate a ResNext-101 (32x8d) pretrained with weakly supervised learning (WSL) on billions of Instagram images (Mahajan et al., 2018), and we obtained the parameters from `https://download.pytorch.org/models/ig_resnext101_32x8-c38310e5.pth`. For the WSL model, we did not use single point BN adaptation as we found this technique to be actually harmful to performance, and this corroborates previous findings (Schneider et al., 2020).

Table 6 summarizes the results. We can see that, similar to Table 2, Both TTA and MEME significantly improve upon the baseline model evaluation. TTA performs best for the baseline ResNext-101 model. However, MEME ultimately achieves the best accuracy by a significant margin, as it is more successful at adapting the WSL model, which has a much higher accuracy. This suggests that combining MEME with other large pretrained models may be an interesting direction for future work.

## C  FULL CIFAR-10-C AND IMAGENET-C RESULTS

In the following tables, we present test results broken down by corruption and level for CIFAR-10-C for the methods evaluated in Table 1. We omit joint training and TTT because these results are available from Sun et al. (2020). Our test results for ImageNet-C are provided in the CSV files in the supplementary material.

Table 7: Test error (%) on CIFAR-10-C level 5 corruptions.

|  | gauss | shot | impul | defoc | glass | motn | zoom | snow | frost | fog | brit | contr | elast | pixel | jpeg |
|---|---|---|---|---|---|---|---|---|---|---|---|---|---|---|---|
| ResNet-26 | 48.4 | 44.8 | 50.3 | 24.1 | 47.7 | 24.5 | 24.1 | 24.1 | 33.1 | 28.0 | 14.1 | 29.7 | 25.6 | 43.7 | 28.3 |
| + TTA | 43.4 | 39.6 | 42.9 | 28.3 | 44.7 | 26.3 | 26.3 | 21.4 | 28.5 | 23.3 | 12.1 | 32.9 | 21.7 | 43.2 | 21.7 |
| + MEME (ours) | 43.5 | 39.8 | 43.3 | 26.4 | 44.4 | 25.1 | 25.0 | 20.9 | 28.3 | 22.8 | 11.9 | 28.3 | 21.1 | 42.8 | 21.7 |

Table 8: Test error (%) on CIFAR-10-C level 4 corruptions.

|  | gauss | shot | impul | defoc | glass | motn | zoom | snow | frost | fog | brit | contr | elast | pixel | jpeg |
|---|---|---|---|---|---|---|---|---|---|---|---|---|---|---|---|
| ResNet-26 | 43.8 | 37.2 | 39.3 | 14.8 | 48.0 | 19.9 | 18.7 | 22.0 | 24.9 | 15.1 | 11.4 | 16.8 | 19.1 | 27.9 | 24.9 |
| + TTA | 39.5 | 32.0 | 31.8 | 15.4 | 45.0 | 20.9 | 20.2 | 18.9 | 21.7 | 12.9 | 9.3 | 16.8 | 17.7 | 25.7 | 18.9 |
| + MEME (ours) | 39.7 | 32.3 | 32.2 | 14.7 | 45.0 | 20.0 | 19.2 | 18.7 | 21.1 | 12.5 | 9.3 | 15.2 | 16.9 | 25.2 | 18.9 |

Table 9: Test error (%) on CIFAR-10-C level 3 corruptions.

|  | gauss | shot | impul | defoc | glass | motn | zoom | snow | frost | fog | brit | contr | elast | pixel | jpeg |
|---|---|---|---|---|---|---|---|---|---|---|---|---|---|---|---|
| ResNet-26 | 40.0 | 33.8 | 26.4 | 11.5 | 37.3 | 20.0 | 16.6 | 20.0 | 24.7 | 12.2 | 10.5 | 13.6 | 15.0 | 18.4 | 22.7 |
| + TTA | 34.3 | 27.7 | 20.3 | 11.3 | 32.9 | 20.7 | 16.7 | 16.3 | 21.1 | 9.8 | 8.5 | 12.5 | 13.7 | 14.5 | 17.2 |
| + MEME (ours) | 34.4 | 27.9 | 20.5 | 10.8 | 32.8 | 19.8 | 16.1 | 16.1 | 20.9 | 9.6 | 8.6 | 11.7 | 13.2 | 14.5 | 17.2 |

Table 10: Test error (%) on CIFAR-10-C level 2 corruptions.

|  | gauss | shot | impul | defoc | glass | motn | zoom | snow | frost | fog | brit | contr | elast | pixel | jpeg |
|---|---|---|---|---|---|---|---|---|---|---|---|---|---|---|---|
| ResNet-26 | 30.1 | 21.8 | 21.1 | 9.7 | 38.3 | 15.3 | 13.8 | 21.2 | 17.6 | 10.5 | 9.7 | 11.6 | 12.9 | 15.4 | 21.3 |
| + TTA | 25.3 | 16.9 | 15.8 | 8.5 | 33.8 | 15.3 | 13.7 | 17.8 | 14.5 | 8.7 | 7.8 | 10.0 | 11.0 | 12.0 | 16.1 |
| + MEME (ours) | 25.3 | 16.9 | 15.9 | 8.4 | 33.5 | 14.6 | 13.0 | 17.7 | 14.4 | 8.5 | 7.7 | 9.6 | 10.7 | 11.9 | 16.2 |

Table 11: Test error (%) on CIFAR-10-C level 1 corruptions.

|  | gauss | shot | impul | defoc | glass | motn | zoom | snow | frost | fog | brit | contr | elast | pixel | jpeg |
|---|---|---|---|---|---|---|---|---|---|---|---|---|---|---|---|
| ResNet-26 | 20.8 | 16.5 | 15.8 | 9.2 | 38.9 | 11.8 | 12.8 | 13.9 | 13.4 | 9.7 | 9.4 | 9.6 | 13.1 | 12.0 | 16.4 |
| + TTA | 15.8 | 12.8 | 11.8 | 7.3 | 35.1 | 10.8 | 12.5 | 11.0 | 10.8 | 7.4 | 7.4 | 7.7 | 11.4 | 9.2 | 12.5 |
| + MEME (ours) | 16.1 | 12.9 | 11.9 | 7.4 | 34.7 | 10.4 | 12.1 | 11.0 | 10.7 | 7.4 | 7.3 | 7.5 | 10.9 | 9.2 | 12.5 |

