# OpenReview forum: "Test Time Robustification of Deep Models via Adaptation and Augmentation"
_ICLR.cc/2022/Conference — ICLR 2022 Submitted_

### Official Review · Reviewer_8PMD · 2021-10-21

**Correctness:** 3
**Technical Novelty And Significance:** 2
**Empirical Novelty And Significance:** 2
**Recommendation:** 5
**Confidence:** 4

**Main Review:**

Strength:
 * The problem of addressing domain shift is a highly significant topic
 * The proposed method can do adaptation on a per-datapoint level (in contrast to related works that require larger batches of inputs from the same target domain)
 * The experiments compare to the most related work on a large set of domain shift benchmarks and baseline models. Results are well-structured
 * The paper is very well written and structured.

Weaknesses:
 * The proposed method has little novelty since it is a straightforward combination of existing methods: test-time augmentation, test-time entropy minimization (Wang et al., 2021), and adapting BatchNorm statistics (Schneider et al. 2020).
 * The proposed method considerably increases inference cost (by a factor of B) - this is undesirable in practice
 * As discussed below, some ablations/alternative design choices are missing

Some alternative design choice that could strengthen the paper when included in the experiments/ablation studies:
 * Equation 1 computes the average of class predictions over augmented samples. Alternatively, the average over logits could be considered.
 * Varying the number of augmentations B in the experiments and discussing the trade-off between inference time and error
 * in general, repeating every run several times with different random seeds and report average error in the results
 * discuss why there is a "stop-gradient" in the loss $\ell_{PCE}$ and study its effect
 *  $\ell_{PCE}$ is not invariant to permutation of the batch since the second sum runs only over $j=(i+1)...B$. Moreover, this is particularly problematic since the gradient flows through the $\tilde{x}_j$-inference for j-1 terms (those terms with i < j) .Thus samples with higher index j have larger weight in the loss. A simple fix would be to use a sum over $j \neq i$.
 * since the authors use AugMix augmentations and AugMix-pretrained models, it would be natural to also add a JSD-type term to the loss at test-time that explicitly aims at encouraging invariance against augmentations.

Minor comments:
 * In the tables, it is confusing to report errors and then a reduction of the error by X percent point with +X (it should be -X since it is a reduction of error)
 * how many iterations of SGD/ADAM are performed per datapoint? I did not find this information in the paper. Is it only one? If yes: why?

**Summary Of The Paper:**

The paper aims at increasing robustness of image classifiers against domain shift using test-time adaptation on a single-instance level. For this, the authors apply a large set of random augmentation on the respective test input and adapt the model using entropy minimization on the marginal distribution of model outputs across augmentations. The proposed method is called "Marginal Entropy Minimization with Ensembled Augmentations" (MEME). MEME is combined with prior methods for compensating domain shift and evaluated on several domain shift benchmarks.

**Summary Of The Review:**

The main contribution of the paper, the method MEME consists of plugging together two existing techniques, namely test-time augmentation and test-time entropy minimization (Wang et al., 2021). This combination is straight-forward and the technical novelty of the paper is quite small.
On the other hand, the empirical evaluation is relatively extensive and well structured . Consistent albeit small gains of the proposed method are observed. As discussed above, there are some ablations/alternative design choice that could be added to the experiments. Including those would give  a more complete picture.

In summary, I see this paper as marginally below the decision threshold, due to its limited novelty and incomplete ablations.

---

> ### Author Response · Authors · 2021-11-21
> **Response to reviewer 8PMD**
>
> Thank you for your helpful comments and suggestions. As requested, we have run the following experiments: studying the efficiency vs accuracy tradeoff when varying B, and evaluating different versions of the ablation involving \ell_{PCE}. We have added these results to the paper and summarize them here. We also wish to respond to your concerns about novelty and address your other minor comments.
>
> > “MEME consists of plugging together two existing techniques, namely test-time augmentation and test-time entropy minimization”
> > “it would be natural to also add a JSD-type term to the loss at test-time that explicitly aims at encouraging invariance against augmentations.”
>
> We wish to clarify that MEME is not just the combination of test time augmentation and entropy minimization. In fact, the MEME objective already explicitly encourages invariance against augmentations during adaptation, since the marginal entropy across augmentations is minimized when the model outputs the same prediction regardless of the augmentation. This is the difference between the MEME objective and the ablation which uses the \ell_{CE} objective, which would be the straightforward version of “plugging together two existing techniques”. We have clarified this distinction in Section 3.1.
>
> To further emphasize this point, we have updated Table 3 with additional results for the confidence maximization ablation (which adapts using \ell_{CE}) on the ImageNet test sets, focusing on the robust vision transformer model. These results show that confidence maximization also helps for the ImageNet problems (e.g., -8.2 mCE over the baseline on ImageNet-C), but MEME still performs better on ImageNet-C (-8.8 mCE) and ImageNet-R. The consistency with which MEME outperforms simple confidence maximization is significant and demonstrates that additionally encouraging invariance to augmentations leads to a more successful algorithm.
>
> |  | ImageNet-C mCE | ImageNet-R Error (%) | ImageNet-A Error (%) |
> | :--- | ---: | ---: | ---: |
> | RVT*-small | 49.4 | 52.3 | 73.9 |
> | MEME (ours) | **40.6** | **43.8** | 69.8 |
> | Confidence maximization | 41.2 | 44.2 | **69.7** |
>
> > “Varying the number of augmentations B in the experiments and discussing the trade-off between inference time and error”
>
> As requested, we have added in Appendix B.2 (Figure 3) an experiment on ImageNet-R in which we vary the number of augmentations B from [1, 2, 4, 8, …, 128] and plot seconds per evaluation (x axis) against test error (y axis). We notice that significant accuracy gains are obtained for as few as 4 augmentations, indicating that a practical tradeoff between efficiency and accuracy is possible. For large B, the wall clock time is dominated by computing the augmentations -- in our implementation, we do not compute augmentations in parallel, though in principle this is possible for AugMix and should improve efficiency overall.
>
> > “discuss why there is a ‘stop-gradient’ in the loss \ell_{PCE} and study its effect”
> > “\ell_{PCE} is not invariant to permutation of the batch since the second sum runs only over j = (i+1) \ldots B… A simple fix would be to use a sum over j \neq i.”
>
> Thank you for pointing out this potential issue. We have fixed the \ell_{PCE} ablation to use a sum over j \neq i, and we have rerun this ablation and updated the results in Table 3. This fix actually made the results on the original CIFAR-10 test set and the CIFAR-10.1 test set somewhat worse, but it significantly improved the CIFAR-10-C results, which are now on par with the other adaptation ablation (\ell_{CE}). Though we found the stop gradient operator to be important for the previous formulation of \ell_{PCE}, we now find for this new formulation that it does not change the results significantly and thus we have removed it. These changes are reflected in the equation for \ell_{PCE} in the revised paper.
>
> > “In the tables, it is confusing to report errors and then a reduction of the error by X percent point with +X”
>
> Thank you for pointing out this confusion, we have fixed the tables by swapping the “+” and “-” symbols.
>
> > “how many iterations of SGD/ADAM are performed per datapoint? I did not find this information in the paper. Is it only one? If yes: why?”
>
> We perform only one iteration per data point because we found it sufficient for improved performance and it improves the computational efficiency of inference. We have clarified this point in Section 3.1.
>
> Thank you for helping to improve the paper! We believe that this response should address all of your points, but please let us know if you have any additional questions or concerns.

---

> > ### Comment · Reviewer_8PMD · 2021-11-22
> > **Feedback to author response**
> >
> > I would like to thank the authors for taking my comments into account and revising the manuscript. Revising \ell_{PCE} and the experiments on the number of augmentations are very helpful.
> >
> > However, I will stick to my general assessment because:
> >  * Novelty is quite small. Test-time augmentation and entropy minimization are prior work; there is just one free choice when combining them: whether to first marginalize over predictions and then compute entropy (MEME) or first compute entropies and then average over those (\ell_{CE}). Choosing MEME over \ell_{CE} makes sense, but it is not very novel.
> >  * That MEME "encourages invariance against augmentations" is not well substantiated. There is not analysis that shows for instance that JSD between predictions on different augmentations decreases more for MEME than for, e.g., \ell_{CE}. Or alternatively, one could study a distance in logit space between the network outputs for different augmentations of the same input.
> >  * Related to this: these metrics to measure invariance against augmentations could also be used as additional loss terms. Studying these options would give a more complete picture for the potential of test-time augmentation and the connection between invariance against augmentation and test-time adaptation.

---

> ### Author Response · Authors · 2021-11-25
> **Followup response to reviewer 8PMD**
>
> Thank you for your response! We are glad to hear that our first response addressed your initial comments. We hope to continue this useful discussion and respond to your remaining points.
>
> > “Novelty is quite small… Choosing MEME over \ell_{CE} makes sense, but it is not very novel.”
>
> Thank you, your clarification is quite helpful in understanding how the work may be perceived. While we agree that MEME is relatively straightforward from a technical perspective, we hope that you can consider other axes of novelty that we feel are significant:
>
> 1. Novelty in the problem setup. Prior approaches for test time adaptation focus on having access to many test points and only tangentially study the robustness (single test point) setting, if at all. On the other hand, papers that have considered the single test point setting have not studied how adapting the model may result in better robustness. We believe that MEME is of interest to researchers in both of these areas, as the robustness test setting is important and our adaptation based approach to this setting is novel.
>
> 2. Novelty in the empirical results. Simply put, MEME achieves better results across a range of models on several challenging benchmarks. We note that, with ICLR’s new guidelines on reviewing for novelty beyond “technical novelty”, the “empirical novelty” category includes both “advancements” and “insights”. We believe that improving results on these benchmarks constitutes significant advancements, and related to the first point, MEME is based on the important insight that adaptation techniques should be studied for the robustness setting.
>
> > “There is not analysis that shows for instance that JSD between predictions on different augmentations decreases more for MEME than for, e.g., \ell_{CE}.”
>
> This is a great suggestion which we have now run for the robust vision transformer model on the ImageNet-R and ImageNet-A test sets. The results are summarized in the table below. We can see that MEME does result in lower average JSD between predictions on different augmentations than \ell_{CE}, thus better substantiating the claim that MEME’s marginal entropy objective encourages greater invariance to augmentations.
>
> | Average JSD between predictions  | ImageNet-R | ImageNet-A |
> | :--- | ---: | ---: |
> | MEME (ours) | 0.173 | 0.230 |
> | Confidence maximization | 0.178 | 0.237 |
>
> This result makes sense and is the consequence of, as you put it, MEME marginalizing first and then computing entropy rather than computing entropies and then averaging. With the latter approach, the model can learn to output confident, but different, predictions on different augmentations, whereas this is penalized by the marginal entropy (MEME) loss. We will add this result to the revised version of the paper.
>
> We hope that these additional comments and results can address your remaining concerns, and we look forward to hearing back from you!

---

> > ### Comment · Reviewer_8PMD · 2021-11-25
> > **Reply to follow-up response**
> >
> > I would like to thank the authors for the additional experiment on the JSD between predictions. This goes in a promising direction; the change in JSD seems to be relatively small and it is unclear how systematic this decrease is, so further empirical evidence to strengthen this point would be desirable.
> >
> > Regarding novelty:
> >  * Novelty in the problem setup: as the authors have written themselves, prior work on test-time adaptation has tangentially studied the robustness (single test point) setting, albeit not as the core setting. It is good that this N=1 test-time adaptation setting gets more attention, but also again it is not novel on its own.
> >  * "Novelty in the empirical results": I was not able to find the part of the reviewer guidelines the authors were referring to (“technical novelty” versus “empirical novelty”) under https://iclr.cc/Conferences/2022/ReviewerGuide . Could the authors point me to the page they refer to? In any case, I would expect this to refer to empirical results that are somewhat unexpected and not incremental. I do not consider this to be the case in this work - combining TTA, TENT, and Single point BN was likely to improve upon each of these parts when applied in isolation.
> >
> > In summary, I keep my general assessment of existing but also small novelty for this work.

---

### Official Review · Reviewer_ZgMY · 2021-10-22

**Correctness:** 3
**Technical Novelty And Significance:** 4
**Empirical Novelty And Significance:** 4
**Recommendation:** 8
**Confidence:** 5

**Main Review:**

I think that the general idea is interesting. It is indeed a short-coming of BatchNorm adaptation that one needs large batch sizes for this technique to really be effective. It is very relevant to have a method that works for single images. This would allow adapting models in an online-adaptation setting where there is only one image present from a certain distribution shift.

The authors should explicitly define the difference between f and p since currently, they seem to be using both interchangeably. Is p just f+softmax? Based on algorithm 1, I understand the procedure as following: The authors use one single test image and create B augmentations of it, pass B through the network, get B embeddings, calculate the mean over the embeddings, then calculate the loss based on the mean, i.e. for calculating the loss, they use p_tilde which is one number per dimension. The difference to the second ablation is that in the second ablation, they average after calculating H. Please make the distinction clearer in the text.

I appreciate the authors uploading their code. I actually went through it to better understand the marginal entropy loss function and several other things.

Comparison to TENT. The authors write “This ablation is effectively a version of Tent (Wang et al., 2021) that is suitable for our test setting.” To me, the most important difference to TENT is that the model is reset after each image while in TENT, the model continually adapts to the full ImageNet-C set. Therefore, ablation 2 is NOT equivalent to TENT and this should be made clear.

I think that the results are rather limited and several important ablations are missing. Especially, for a purely experimental paper I think more results should be presented.
1.	The most relevant difference to other gradient-based test-time approaches is that the model is reset after each new image. It has not been tested whether TENT still works if one resets the model after each batch.
2.	It is not clear to me how many gradient steps are performed on each image. In the code, n_iter is set to one but I think it is a hyperparameter that should be tuned.
3.	How is B selected? I would appreciate a study for different values of B. In the limit case of B=1, one recovers the TENT setting, right? Currently, it is not clear to me that TENT does not work for B=1 since it has not been shown and TENT (with model resetting) actually performs equally well to MEME in Table 2 for a batch size of 64. -> Showing that TENT (+ model resetting) indeed does not work for small batch sizes and one needs this averaging over augmentations would be important.
4.	Currently, the authors only study ResNet50 models and the Vision Transformer. For the ImageNet-A results, any results on ResNet50 models are biased (see my points below) and it is not clear how to adapt vision transformers (see my points below), making the results on both architectures hard to judge. I would suggest to the authors to test other model architectures, e.g., they could test the ResNext101 models (both vanilla and the IG-3.5B variants), which would be interesting because the IG-3.5B variant could not be adapted successfully with Batch Norm adaptation.
4.b. In the abstract, the authors write "In our experiments, we demonstrate that this approach consistently improves robust ResNet and vision transformer models". To avoid over-claiming the success of the method, I would suggest to the authors to include all relevant robust ResNet50 models, e.g., the ones from Table 1 in Schneider et al. Alternatively, the authors have to remove the claim that MEME improves upon all robust ResNet50 models and write that MEME improves upon the vanilla, the DeepAug+Augmix and the MoEx+CutMix model.
5.	Considering the second ablation point: “Conversely, as an objective that encourages confidence but not invariance…”, I find the presented results not conclusive enough to state that both confidence and invariance are important, as the difference are only 0.4 percent max. Possibly, more experiments should be performed to really stress this point.
6.	AugMix augmentations come in different severity levels. Checking the code, it seems the authors used a severity of 1 which is the smallest severity possible. This might explain the ablation result why the difference to standard augmentations is not that big. I think varying this parameter would be important.
7. I would find it very interesting to see how MEME performs for larger batch sizes. The authors could plot a curve like Figure 1 in Schneider et al to see the effect of different batch sizes (alongside with the BN adapt curve). Ideally, the MEME curve should always be below the "BN adapt" curve. In the limes of the full dataset size as sample size, MEME should converge to the TENT result. Using the full dataset size as the batch size is obviously not possible due to GPU constraints, but one should see the asymptotic trend approaching that number. Maybe the authors could use batch sizes of [1,2,4,8,16... max_batch_size_per_GPU] for this analysis.

### Comments on achieving a new state of the art claims:
The authors claim to achieve a new state of the art on various benchmarks on several occasions.
1. “We achieve state-of-the-art results for test shifts caused by image corruptions (ImageNet-C), renditions of common objects (ImageNet-R), and, among ResNet-50 models, adversarially chosen natural examples (ImageNet-A).”
2. “In Section 4, we demonstrate empirically that MEME consistently improves the performance of ResNet (He et al., 2016) and vision transformer (Dosovitskiy et al., 2021) models on several challenging ImageNet distribution shift benchmarks, achieving several new state-of-the-art results for these models.”
3. “In particular, as we empirically show in Section 4, MEME can be composed with prior methods for training robust models and adapting model statistics, resulting in improvements over the prior state of the art for several ImageNet distribution shift benchmarks and model architectures.”
4. “For both ImageNet-C and ImageNet-R, and for both the ResNet-50 and RVT_-small models, combining MEME with robust training techniques leads to new state of the art performance.”

When speaking about state-of-the-art results on either ImageNet-C or ImageNet-R, one can either claim (1) a new state of the art for a certain architecture or (2) a new state of the art on ImageNet-C overall.
Considering case (1), the authors claim state-of-the-art results on ImageNet-C for a ResNet50 architecture which is wrong since both TENT and BN adaptation outperform their approach.
Considering case (2), the authors report a mCE of 40.6% on ImageNet-C and an error of 43.8% on ImageNet-R for their vision transformer. These are not state of the art overall, because for example, the Noisy Student model achieves a mCE of 28.3% on ImageNet-C and an error of 23.5% on ImageNet-R.

The authors achieve good results for the vision transformer model, but this is not a standard architecture (yet) so judging these numbers is hard. It is in general not very clear how transformers should be adapted. In conv-nets, usually the batchnorm layers are adapted (like in TENT), but maybe in transformers, one needs to adapt other parameters. For example, [5] and [6] experiment with “adapter” and “compacter” layers, respectively, to adapt transformers. The authors also raise this point when discussing their results: “Furthermore, we find that these methods are less effective with the RVT-small model, which may indicate their sensitivity to model architecture choices since vision transformer models are comparatively less reliant on BN layers (Steiner et al., 2021).” Yes, exactly. But to me, it does not mean that TENT and BN adaptation do not work per se, but that one needs to think about which layers need to be tuned.
-> It might be that TENT would work if one used the layers suggested there, but not if one uses conventional BN adaptation.

On ImageNet-A, the Noisy Student model from [3] reports a top1 error of 16.5% which is much better than what the authors report here. Sure, the authors manage to use MEME to decrease the top1 error of a ResNet50 backbone to 89% (which is a slight improvement from 92%), but I find this result not very interesting, because it is still very bad and no one would actually use this model on ImageNet-A. In either case, their ImageNet-A results are not state of the art. In general, all ResNet50 results on ImageNet-A are very biased, because this dataset was explicitly designed to fool vanilla ResNet50 classifiers. So results reported for a ResNet50 do not generalize to other architectures. One can argue that MEME is the only method that improves upon the RVT results, but here again the issue from above arises where it is not clear how to adapt a vision transformer properly.

Table 2: “A composition of robust training via strong data augmentations and MEME adaptation achieves new state-of-the-art performance on each benchmark for ResNet-50 models.” -> This is wrong: DeepAugment+Augmix+TENT achieve lower error rates on ImageNet-R and ImageNet-C for the ResNet50 backbone. I also find the bolding of the numbers inconsistent: If the best numbers are supposed to be bold, why is MEME bold as well despite being worse than TENT? The authors write “and we list in bold
the best results from these methods which outperform the test time robustness methods.” I don’t get it: BN adaptation and TENT are also test time robustness methods?


As a summary of this point, I think the authors need to rephrase their interpretation of the results since they do not achieve SotA results neither on ImageNet-C nor on ImageNet-R nor on ImageNet-A.


**Summary Of The Paper:**

The authors study test-time adaptation to distribution shifts. They address the problem of single-sample adaptation and propose a method where they create several augmentations of a single image and average over the augmentations. They present results on different common robustness benchmarks.

**Summary Of The Review:**

I really like the general idea of the paper of trying to find a method where adaptation to a single test sample is possible, with resetting the model after each sample. To my knowledge, this has not been done yet and it would be an interesting contribution to the field and highly relevant for practitioners. This is actually very similar to how robustness is evaluated when model weights are frozen and inference is done on every point separately. Here, one wants to adapt to each test data point.

It is obviously a harder problem since so far, models have been allowed to adapt to the whole test set, e.g. TENT studied this setting. And indeed, single-point BN adaptation is currently the setting closest to this one. So it is fine to not claim state-of-the-art results on ImageNet-C in this hard setting. Of course, methods that are allowed to see the full test set and adapt to it will be better. Therefore, I think the authors should refrain from wanting to claim state-of-the-art results over methods that are allowed to see the full test set, since it is an unreasonable request. Methods that are allowed to see the full test set should always perform better, and if they don’t, there is likely an optimization issue like with the vision transformers.

I would suggest to the authors to pitch their paper in this setting: It is a new problem that no one has really studied so far which we propose a solution for and we ran all the necessary control experiments. That said, if one has access to many samples from the target domain, it should still be better to use all of them for adaptation. I think writing the pitch in this way should make the paper much more clear and easier to understand and put into context.

If one were to pitch the paper in this setting, one would need to test whether other test time adaptation methods work with model resetting, i.e. does TENT work for a batch size of 1 and with model resetting? Does one really need the different augmentations as claimed here?

I think that the paper cannot be accepted in the current state, but I think the general setting is very interesting and it can become a relevant contribution if the pitch is changed and more results are added. I am open to have a discussion with the authors during the rebuttal period whenever they wish to. I am happy to raise my score if my points are addressed.

---

> ### Author Response · Authors · 2021-11-14
> **Initial Response and a Question for Reviewer ZgMY**
>
> Thank you for your extensive and detailed comments. In the coming days, we will follow up with a full response, including the requested experiments. But we want to first follow up to resolve your concerns around the framing of “state of the art” and clarify a potential misunderstanding.
>
> > “I think the authors need to rephrase their interpretation of the results since they do not achieve SotA results”
>
> Upon further review, we agree that the paper’s state-of-the-art claims can be misleading, and we thank you for pointing this out. Our intent was to claim state-of-the-art results “for a certain architecture”, as you have suggested, *and for a certain test setting*. When we refer to “our test setting”, we mean the setting in which only one test point is available, i.e., the standard evaluation setting. This is why we do not consider the Tent and BN adaptation results in this claim and color their results in gray. Using the definition at the beginning of Section 3, these methods are not test time robustness methods (though other prior methods like TTA and single point BN adaptation are). **We have revised the paper such that all mentions of “state of the art” are further qualified with “in the setting where only one test point is available” -- does this resolve your concern about the state-of-the-art claims?**
>
> > “TENT (with model resetting) actually performs equally well to MEME in Table 2 for a batch size of 64”
>
> Potential misunderstanding: we wish to note that the Tent results in Table 2 are *not with model resetting*: the Tent model continually adapts while predicting over the entire test set, so by the end of evaluation, the Tent model has the benefit of having seen *all of the test points*. We have clarified this in Section 4, and we will also provide results for Tent + model resetting soon.

---

> > ### Comment · Reviewer_ZgMY · 2021-11-19
> > **Response to the authors' comments**
> >
> > Thank you for addressing my comment on the state-of-the-art claims. The current formulation is fine and exactly what I had in mind.
> >
> > Considering the second comment, yes, I agree. I apologize for the confusion: I meant Table 3 and not Table 2. In Table 2, TENT performs better than MEME, while in Table 3, both are almost equal. Table 3 is done with model resetting, right? So TENT works for a batch size of 32 with model resetting?

---

> ### Author Response · Authors · 2021-11-21
> **Followup response to reviewer ZgMY (part 1)**
>
> Thank you for your response! We are glad that our paper revision has resolved your concerns around the “state-of-the-art” framing. We are following up with additional experiments as requested, which we believe should address your remaining concerns. This response is split up into two parts, due to character limits.
>
> > “Showing that TENT (+ model resetting) indeed does not work for small batch sizes and one needs this averaging over augmentations would be important.”
> > “does TENT work for a batch size of 1 and with model resetting? Does one really need the different augmentations as claimed here?”
>
> We have added experiments in Appendix B.1 (Table 4) in which we evaluate Tent with a batch size of 1 with model resetting. As you point out, this is analogous to MEME with B=1. Tent typically uses BN adaptation on the batch of test points to update \mu and \sigma^2. Also analogously to MEME, as described in Section 3.2, here we use a prior strength of N=16 for Tent’s BN adaptation, to prevent overfitting to the single test point. These results show that this method does not provide any additional performance gains over just using the single point BN adaptation. This indicates that using augmentations is important for achieving the reported results. Table 4 is too large to reproduce here, but as an example, here are the baseline ResNet-50 results:
>
> |  | ImageNet-C mCE | ImageNet-R Error (%) | ImageNet-A Error (%) |
> | :--- | ---: | ---: | ---: |
> | Baseline ResNet-50 | 76.7 | 63.9 | 100.0 |
> | Single point BN | 71.4 | 61.1 | 99.4 |
> | MEME (ours) | **69.9** | **58.8** | **99.1** |
> | Tent (batch size 1, with model resetting) | 71.4 | 61.2 | 99.4 |
>
> > “To me, the most important difference to TENT is that the model is reset after each image while in TENT, the model continually adapts to the full ImageNet-C set.”
> > “It has not been tested whether TENT still works if one resets the model after each batch.”
>
> We have now added results to Table 2 for the ResNet models and an episodic version of Tent which resets the model after each batch of 64 images. We note that this method still performs reasonably well as it is given 64 images for adaptation, rather than just one, but it generally, and predictably, performs worse than the online version of Tent which adapts on the entire test set. Again, as an example, here are the baseline ResNet-50 results, with the full results in Table 2:
>
> |  | ImageNet-C mCE | ImageNet-R Error (%) | ImageNet-A Error (%) |
> | :--- | ---: | ---: | ---: |
> | Baseline ResNet-50 | 76.7 | 63.9 | 100.0 |
> | Tent (online) | **54.4** | **57.7** | 99.8 |
> | Tent (episodic) | 64.7 | 61.0 | **99.7** |
>
> > “How is B selected? I would appreciate a study for different values of B.”
>
> We selected B=64 to mimic the batch size used in Tent. We have added in Appendix B.2 (Figure 3) an experiment on ImageNet-R in which we vary the number of augmentations B from [1, 2, 4, 8, …, 128] and plot seconds per evaluation (x axis) against test error (y axis). We notice that significant accuracy gains are obtained for as few as 4 augmentations, indicating that a practical tradeoff between efficiency and accuracy is possible. For large B, the wall clock time is dominated by computing the augmentations -- in our implementation, we do not compute augmentations in parallel, though in principle this is possible for AugMix and should improve efficiency overall.

---

> > ### Author Response · Authors · 2021-11-21
> > **Followup response to reviewer ZgMY (part 2)**
> >
> > Here is the second part of our response.
> >
> > > “For the ImageNet-A results, … I would suggest to the authors to test other model architectures, e.g., they could test the ResNext101 models (both vanilla and the IG-3.5B variants), which would be interesting because the IG-3.5B variant could not be adapted successfully with Batch Norm adaptation.”
> >
> > We have added in Appendix B.3 (Table 6) results for both the vanilla and IG-3.5B ResNext101 (32x8d) models evaluated on ImageNet-A. These results indicate that MEME is also an effective strategy for adapting these models, further highlighting its general applicability. As we have seen in the previous ImageNet-A results, TTA is a strong baseline and performs better for the vanilla model, however, MEME achieves the best accuracy overall by a large margin, as it is more successful at improving the IG-3.5B model.
> >
> > | | ImageNet-A Error (%) |
> > | :--- | ---: |
> > | Baseline ResNext-101 (32x8d) | 90.0 |
> > | TTA | 83.2 |
> > | MEME (ours) | 84.3 |
> > | | |
> > | ResNext-101 (32x8d) IG-3.5B | 54.9 |
> > | TTA | 49.1 |
> > | MEME (ours) | **43.2** |
> >
> > > “I find the presented results not conclusive enough to state that both confidence and invariance are important, as the difference are only 0.4 percent max. Possibly, more experiments should be performed to really stress this point.”
> > > “in Table 3, both [confidence maximization and MEME] are almost equal.”
> >
> > We have updated Table 3 with additional results for the confidence maximization ablation on the ImageNet test sets, focusing on the robust vision transformer model. These results show that confidence maximization also helps for the ImageNet problems (e.g., -8.2 mCE over the baseline on ImageNet-C), but MEME still performs better on ImageNet-C (-8.8 mCE) and ImageNet-R. The consistency with which MEME outperforms simple confidence maximization is significant and demonstrates that additionally encouraging invariance to augmentations leads to a more successful algorithm.
> >
> > |  | ImageNet-C mCE | ImageNet-R Error (%) | ImageNet-A Error (%) |
> > | :--- | ---: | ---: | ---: |
> > | RVT*-small | 49.4 | 52.3 | 73.9 |
> > | MEME (ours) | **40.6** | **43.8** | 69.8 |
> > | Confidence maximization | 41.2 | 44.2 | **69.7** |
> >
> > > “it does not mean that TENT and BN adaptation do not work [for the vision transformer] per se, but that one needs to think about which layers need to be tuned.”
> >
> > We have run a version of Tent for the vision transformer where all parameters are updated and updated Table 2 with the results. We report results from both versions of Tent because we found that, compared to standard Tent adaptation, adapting all parameters improves results for ImageNet-C (-2.1 relative mCE improvement) and ImageNet-A (-1% relative error) but is catastrophic for ImageNet-R (+23.4% relative error). We noticed that this drop was largely due to entropy minimization changing the model too aggressively later on in the evaluation loop, and we believe that this indicates the careful design decisions that must be made when performing continual adaptation. Comparatively, MEME does not require these design decisions as we consider the test time robustness (single test point) setting, and for the simple choice of adapting all parameters, MEME is successful for all three ImageNet datasets.
> >
> > |  | ImageNet-C mCE | ImageNet-R Error (%) | ImageNet-A Error (%) |
> > | :--- | ---: | ---: | ---: |
> > | RVT*-small | 49.4 | 52.3 | 73.9 |
> > | MEME (ours) | **40.6** | **43.8** | **69.8** |
> > | Tent (standard) | 46.8 | 50.7 | 82.1 |
> > | Tent (adapt all) | 44.7 | 74.1 | 81.1 |
> >
> > > “I also find the bolding of the numbers inconsistent: If the best numbers are supposed to be bold, why is MEME bold as well despite being worse than TENT?”
> > > “In Table 2, TENT performs better than MEME”
> >
> > We have added more discussion of this to Section 4. MEME generally outperforms prior methods that also observe only one test point. However, as you have correctly noted, it is outperformed in most cases by methods that get to see many test points -- a batch of 256, in the case of BN, and a batch of 64 along with all previous batches, in the case of Tent. We stratify methods by the assumptions that they make, indicated by the different colors in Table 2, and we believe it is fair to separately bold the best results by stratification.
> >
> > > “In the abstract, … the authors have to remove the claim that MEME improves upon all robust ResNet50 models and write that MEME improves upon the vanilla, the DeepAug+Augmix and the MoEx+CutMix model.”
> >
> > Thank you for pointing this out, we have updated the abstract in the paper to more specifically represent the results we achieve. Note that we cannot update the OpenReview abstract, though we will make sure that this change propagates properly.
> >
> > Again, thank you for taking the time to provide detailed and helpful feedback, and for helping to improve the paper! We believe that this response should address all of your points, but please let us know if you have any additional questions or concerns.

---

> > > ### Comment · Reviewer_ZgMY · 2021-11-23
> > > **Response to the rebuttal**
> > >
> > > I thank the authors for addressing my points! Great job with the rebuttal! I am raising my score to 6 for now. I am happy to raise it further after my other points have been addressed.
> > > ### Hyperparameter choices:
> > > The authors write: “We select hyperparameters using the four disjoint validation corruptions provided with CIFAR-10-C and ImageNet-C (Hendrycks & Dietterich, 2019). As the other benchmarks are only test sets and do not provide validation sets, we use the same hyperparameters found using the corruption validation sets and do not perform any additional tuning.” This is very good and could the authors please list which hyperparameters they chose in this manner, and for which experiments? I have questions on the learning rate for TENT (comment on “TENT with bsz=1 and model resetting)” and for the vision transformers (comment on “adapting certain layers of the VT”) below. Basically, if the learning rate is chosen with this hyperparameter scheme, I do not need to see additional results on both comments because in this case, the learning rates were chosen using a proper validation set and even if they are too low/too high on the target set, this is fine, because proper model selection has been done.
> > >
> > > I would also really appreciate documenting the hyperparameter selection results in the Appendix. Besides making it really clear, which hyperparameters were considered and in which range, it would also make it easy to judge which technique is sensitive to which hyperparameters. I realize that the paper can no longer be updated; I am happy to see the Tables in markdown.
> > > ### TENT with bsz=1 and model resetting.
> > > Thanks for this experiment! This looks quite promising. What I find interesting is that TENT is actually not better than single point BN. You should note this point in the text. Could it be a learning rate issue? Maybe the learning rate for TENT is too small and therefore nothing happens? Did you use the same learning rate for TENT and for MEME? Can I ask for a learning rate ablation on ImageNet-R to limit the number of computations? Did you use the same n_iter parameter for the TENT experiment? Could it be that one would need to perform multiple steps for TENT, while one does not need this for MEME. I apologize for asking for more experiments, but I find the current result inconclusive. If it happens that it is a learning rate issue or that one needs more iterations for TENT to work, this would be another positive result where MEME is more stable than TENT? (Please only perform this experiment if the learning rate and n_iter have NOT been selected following the model selection scheme from above.)
> > > ### Results on the episodic version of TENT
> > > This is a really interesting result! It looks to me like that B=8 is enough and afterwards, the benefits are marginal. You should communicate this, because it reduces the computational complexity and makes the paper stronger.
> > > ### Results on the ResNext101 models
> > > Thanks for doing this experiment! I would suggest to still report the single point BN adaptation result for future reference.  It’s cool that MEME works well for adapting the WSL pretrained model. This should be highlighted in the main text as it further strengthens the paper and the technique.
> > > ### Results on adapting certain layers of the vision transformer
> > > Thank you for doing this experiment! I think it’s an interesting result that the performance of the vision transformer breaks down and seems to depend on the dataset. Could it be due to a too high learning rate? The adaptation mechanism, i.e. which layers should be adapted is a hyperparameter choice; has this been selected using the holdout corruptions?  (Please only perform this experiment if the learning rate and the adaptation mechanism have NOT been selected following the model selection sheme)
> > > ### ImageNet-A results
> > > I find it really peculiar how different models perform on ImageNet-A. The ResNet50 models obviously have random chance performance. Then, the ResNext101 model performs badly if trained on ImageNet and better if trained on IG-3.5B. This makes me think that the issue with ImageNet-A has more to do with the dataset rather than with the architecture. But then, the performance of the vision transformer model is good, although it has been trained on ImageNet? Could the authors share their thoughts on this issue?
> > > ### Additional comments:
> > > My comment 2 has not been addressed I believe: “It is not clear to me how many gradient steps are performed on each image. In the code, n_iter is set to one but I think it is a hyperparameter that should be tuned.”
> > > In principle, I would find it interesting to consider the setting of batch size > 1 for MEME. Basically, if we have access to more images from the same domain, should we do MEME or should we do TENT? At which point does it switch? Thinking from a practitioner’s perspective, I think it is clear now that if we have one image, we should do MEME and not TENT. But what if we have 2 or 64 images?

---

> ### Author Response · Authors · 2021-11-25
> **Second followup response to reviewer ZgMY (part 1)**
>
> Thank you for taking our response into consideration and improving your score! We are eager to continue this conversation in order to further improve the paper by addressing your remaining points. This response is again split up into two parts, due to character limits.
>
> > “could the authors please list which hyperparameters they chose in this manner, and for which experiments?”
> > “I would also really appreciate documenting the hyperparameter selection results in the Appendix.”
>
> We will make sure to include the following details and tables in the revised version of the paper.
>
> We used the corruption validation sets to tune the following hyperparameters in a grid search within the following ranges:
>
> | Hyperparameter | Values considered |
> | :--- | ---: |
> | learning rate | 1e-3, 1e-4, 1e-5, 1e-6; then, 5x, 2.5x, 0.5x the best value |
> | # gradient steps | 1, 2 |
> | threshold | 0.5, 1.0 |
> | prior strength | 8, 16, 32 |
>
> Beyond learning rate and number of gradient steps, we also evaluated using a simple “threshold” by performing adaptation only when the marginal entropy was greater than 0.5 of the maximum value (log(1000) for ImageNet-C), though we found that this resulted in slightly worse validation performance. As now described in greater detail in Section 5, more sophisticated techniques for selective adaptation would be interesting to study for future work. We also considered different values of the prior strength N for single point BN adaptation, varying from 16 which is the value suggested in prior work, and we found that 16 performed best on the validation set.
>
> Note that we did not initially sweep over B. Now that we have run this study (Figure 3 in the paper) thanks to the helpful suggestions from you and other reviewers, we agree that smaller values of B, such as 8, provide a favorable efficiency to performance tradeoff, and we will make sure to communicate this in the revised main paper.
>
> For the robust vision transformer, as this is presently a less standard model, after first tuning the hyperparameters above, we further tuned the following hyperparameters in a grid search within the following ranges:
>
> | Hyperparameter | Value considered |
> | :--- | ---: |
> | optimizer | SGD, Adam |
> | weight decay | 0.0, 0.1 |
> | prior strength | 1, 2, 4, 8, 16, 32, $\infty$ (i.e., no single point BN adaptation) |
>
> > “TENT with bsz=1 and model resetting… Could it be a learning rate issue? … Could it be that one would need to perform multiple steps for TENT, while one does not need this for MEME.”
>
> For this version of Tent, which is analogous to a version of MEME with B=1, we did employ the same learning rate sweep on the corruption validation sets as detailed above when we reported the results in Table 4. We previously did not sweep the number of gradient steps, thank you for this suggestion! We have done so now and do indeed find improved performance for this version of Tent with two gradient steps. We include the whole table here, as we cannot update the paper, but these results will be included in a revised version of the paper:
>
> |  | ImageNet-C mCE | ImageNet-R Error (%) | ImageNet-A Error (%) |
> | :--- | ---: | ---: | ---: |
> | Baseline ResNet-50 | 76.7 | 63.9 | 100.0 |
> | Single point BN | 71.4 | 61.1 | 99.4 |
> | MEME (ours) | 69.9 | 58.8 | 99.1 |
> | Tent (batch size 1, with model resetting) | 70.4 | 60.0 | 99.3 |
> | | | | |
> | DeepAugment + AugMix | 53.6 | 53.2 | 96.1 |
> | Single point BN | 51.3 | 51.2 | 95.4 |
> | MEME (ours) | **49.8** | **49.2** | 94.8 |
> | Tent (batch size 1, with model resetting) | 50.7 | 50.7 | 95.2 |
> | | | | |
> | MoEx + CutMix | 74.8 | 64.5 | 91.9 |
> | Single point BN | 71.0 | 62.6 | 91.1 |
> | MEME (ours) | 69.1 | 59.4 | **89.0** |
> | Tent (batch size 1, with model resetting) | 69.9 | 61.7 | 90.6 |
> | | | | |
> | | | | |
> | RVT*-small | 49.4 | 52.3 | 73.9 |
> | Single point BN | 48.0 | 51.1 | 74.4 |
> | MEME (ours) | **40.6** | **43.8** | **69.8** |
> | Tent (batch size 1, with model resetting) | 47.9 | 50.9 | 74.4 |
>
> We note that, although this version of Tent is now generally better than single point BN adaptation, it still lags behind MEME. MEME uses one gradient step and generally does not seem particularly sensitive to this choice, which, as you note, is a positive result for MEME. Taking two gradient steps also effectively doubles the runtime of the method.
>
> > “I would suggest to still report the single point BN adaptation result [for the ResNext-101 models] for future reference.”
>
> We will make sure to update the paper with this result. Here is the updated table, for reference:
>
> | | ImageNet-A Error (%) |
> | :--- | ---: |
> | Baseline ResNext-101 (32x8d) | 90.0 |
> | TTA | 83.2 |
> | Single point BN | 88.8 |
> | MEME (ours) | 84.3 |
> | | |
> | ResNext-101 (32x8d) IG-3.5B | 54.9 |
> | TTA | 49.1 |
> | Single point BN | 58.9 |
> | MEME (ours) | **43.2** |

---

> > ### Author Response · Authors · 2021-11-25
> > **Second followup response to reviewer ZgMY (part 2)**
> >
> > Here is the second part of our response.
> >
> > > “Results on adapting certain layers of the vision transformer… Could it be due to a too high learning rate? … which layers should be adapted is a hyperparameter choice; has this been selected using the holdout corruptions?”
> >
> > For reporting these results, we also reran a sweep of the learning rate and optimizer on the validation data, in the same procedure as above. We did not sweep the weight decay or prior strength hyperparameters. It is certainly possible that this learning rate is too high for ImageNet-R, since the learning rate was tuned using the ImageNet-C validation set. As for sweeping over which layers to adapt, we only considered standard Tent (adapting BN layers) vs adapting all parameters, as in MEME, and we reported both sets of results in Table 2. The search space is rather large for selecting which parameters to adapt, but if you have any suggestions for choices you would like to have evaluated on the validation data, we are happy to try running these by the end of the discussion phase.
> >
> > > “I find it really peculiar how different models perform on ImageNet-A… Could the authors share their thoughts on this issue?”
> >
> > We believe the issue with ImageNet-A is related to a point you raised in your original review: “this dataset was explicitly designed to fool vanilla ResNet50 classifiers” trained on ImageNet. It seems plausible that because the ResNext model architecture is relatively similar to the ResNet architecture, there could be a strong correlation between which data points one would pick to fool these models if they are both trained on ImageNet. On the other hand, vision transformers differ much more drastically in terms of architecture, so this model may be susceptible to a different set of data points, even when trained with the same training data. Thus, it seems likely that changing either of these axes (model architecture or training data) should change the results on ImageNet-A, due to how the test set was constructed.
> >
> > > “It is not clear to me how many gradient steps are performed on each image. In the code, n_iter is set to one but I think it is a hyperparameter that should be tuned.”
> >
> > We apologize for missing this point in our prior response. We did clarify this in Section 3.1 based on another reviewer’s comment. As discussed above, we did not see any significant difference in validation accuracy between using one or two gradient steps, so we stuck with one for efficiency’s sake.
> >
> > > “In principle, I would find it interesting to consider the setting of batch size > 1 for MEME. Basically, if we have access to more images from the same domain, should we do MEME or should we do TENT?”
> >
> > We agree that this is interesting and are happy to include it in Section 5 as a direction for future work. We will also try to run some studies on ImageNet-R by the end of the discussion phase where we compare MEME to Tent when given access to 2, 4, 8, 16, 32, 64, 128, and 256 images.
> >
> > Again, thank you for helping to improve the paper, and please let us know if you have any additional comments!

---

> > > ### Comment · Reviewer_ZgMY · 2021-11-25
> > > **Response to the follow-up comments**
> > >
> > > Thanks for the quick response and the clarifications.
> > >
> > > Thank you for explaining how you did hyperparameter tuning. This seems to be done properly and should be communicated as such in the paper. You should write a short section in the main part where you state which hyperparameters you tuned and in which manner. It can get lost easily if it is only in the appendix. And the question "well maybe they only tuned their own method well and used default hyperparameters for the baseline methods" naturally arises. I had this impression looking at the new results for TENT with model resetting and the VT results. Instead, you should write that you tuned the hyperparameters for all methods in the exact same manner. In this case, something like " It is certainly possible that this learning rate is too high for ImageNet-R, since the learning rate was tuned using the ImageNet-C validation set." is perfectly fine; that's just what happens with model selection. This point is very important to me, as it seems that many papers tune their hyperparameters on the test set and it should really be stressed if the model selection is done properly.
> > >
> > > I am raising my score to 8 now.
> > >
> > > Additional thoughts:
> > > 1. I would try to integrate the new results better into the paper. I know there is a page limit, but they currently feel a bit "stacked" at the end of the paper. I believe they are not even referenced right now.
> > > 2. The paper is centered around MEME, but I would find it more interesting to place it more around the new task. In the sense that, we now have a new task: adaptation to a single test sample, and we now check how different techniques perform. If the authors have to resubmit the paper, I would maybe rewrite it from this perspective. One reviewer writes that "Test-time augmentation and entropy minimization are prior work", and I think the issue here is that it is not as clear that the task is very novel. And for this task, it is absolutely not clear whether we expect entropy minimization to work at all.
> > > I would suggest the following pitch: putting the task into the spotlight and defining it, then showing that single point BN adaptation is the only technique we have for it so far, then showing that conventional entropy minimization does not work better than single point BN adaptation with proper model selection. Then showing how to fix entropy minimization (MEME). For me, the novelty lies in the task and MEME is a way to "fix" entropy minimization. In addition, an empirical evaluation paper on a new task is definitely very valuable for the community. Even if entropy minimization performs better than MEME, it is still interesting, because one simply benchmarks on a new task. In short: From the practitioner's perspective, it is most interesting to know what can be done if one has one single test sample to adapt to.
> > > This comment is rather for the future, in case the authors have to resubmit the paper.

---

### Official Review · Reviewer_6Xgd · 2021-11-01

**Correctness:** 4
**Technical Novelty And Significance:** 2
**Empirical Novelty And Significance:** 2
**Recommendation:** 5
**Confidence:** 5

**Main Review:**

Strengths
1. The proposed method is general since it doesn't need to alter the model training and supports online adaptation with one sample at a time.
2. The single objective embraces two intuitions behind: invariance across augmentations and prediction confidence.
3. The method is simple and easy to implement.

Weaknesses
1. The paper claimed that MEME could adapt only BN statistics in Section 3.2, but there seems no results presented in Section 4's tables.
2. It is unclear how the testing is conducted. Does testing happen after adapting the model over all the test samples (one epoch or several epochs)? Or do we test on a sample immediately after adapting to it? Do different methods use the same testing procedure? More details are necessary at the beginning of the experiment section.
3. According to Table 2, MEME can't beat other methods in several cases. It usually gets higher errors on datasets ImageNet-C and ImageNet-R than BN and Tent. Although it gets better results on ImageNet-A, it still has a higher error than the test-time augmentation baseline.
4. Considering the entropy minimization has been explored in prior work Tent, the main novelty is bringing invariance to augmentations into test-time adaptation.
5. If making model adaptation in deployment, the inference efficiency will suffer from forwarding multiple augmentations and updating the model weights. Any investigation on efficiency? It would be better to add some discussions on how to improve efficiency.

**Summary Of The Paper:**

This paper studies the test-time model adaptation, where training data is not accessible and only one test example is available. It adapts model parameters by minimizing the output entropy on augmented test samples. Given one test sample, it first samples some augmentation operations, resulting in different augmentations. The entropy is computed on the average distribution of augmentations' outputs. Experiments on CIFAR-10 CIFAR-10.1 CIFAR-10-C, ImageNet-C, ImageNet-R, and ImageNet-A show that the proposed method effectively reduces classification errors.

**Summary Of The Review:**

Test-time model adaptation is an interesting problem, providing a new perspective to improve model robustness. The proposed method builds on intuitive assumptions that are easy to understand. There are mainly two concerns regarding novelty and effectiveness. I think the authors may need to rethink designing experiments to outstand advantages.

---

> ### Author Response · Authors · 2021-11-14
> **Initial Response to Reviewer 6Xgd**
>
> Thank you for your helpful comments. We have updated the paper in line with your questions and suggestions, and we summarize the updates here. We believe that this response should address all of your points, but please let us know if you have any additional questions or concerns.
>
> > “The paper claimed that MEME could adapt only BN statistics in Section 3.2, but there seems no results presented in Section 4's tables.”
>
> We have updated Section 3.2 to clarify that we *always* use BN adaptation with MEME, i.e., we adapt the BN statistics alongside all of the model parameters for our method.
>
> > “It is unclear how the testing is conducted.”
>
> We have added more details to the beginning of Section 4 as requested. In short, for MEME, prediction on each test point occurs immediately after adaptation. BN and Tent are colored in gray because they follow a different test protocol as in their respective papers. Specifically, BN predicts on a batch of 256 test points after computing BN statistics on the batch, and Tent predicts on a batch of 64 inputs after adaptation, but it continually adapts throughout the entire test evaluation.
>
> > “According to Table 2, MEME can't beat other methods in several cases.”
>
> We have added more discussion of this to Section 4. MEME generally outperforms prior methods that also observe only one test point. However, it is outperformed in most cases by methods that get to see many test points -- a batch of 256, in the case of BN, and a batch of 64 along with all previous batches, in the case of Tent. TTA is a strong baseline for ImageNet-A in particular, and here we find mixed results for the relative benefits of MEME. As noted, prior adaptation methods have only reported negative results for ImageNet-A.
>
> > “Any investigation on efficiency? It would be better to add some discussions on how to improve efficiency.”
>
> In the coming days, we will add experiments that vary the number of augmentations B and report accuracy as a function of inference time. This will help supplement the discussion of efficiency, which is currently limited to Section 5. We will follow up again with these results.
>
> Thank you for helping to improve the paper!

---

> > ### Comment · Reviewer_6Xgd · 2021-11-20
> > **Response to the authors' comments**
> >
> > Thank you for addressing my comments and clarifying the testing protocols.
> >
> > 1. I am still a little confused about what parameters MEME adapts. Section 3.2 in the revised version says, "That is, we always use single point BN adaptation in conjunction with MEME." However, in the above reply, you wrote, "i.e., we adapt the BN statistics alongside all of the model parameters for our method." In other words, the paper claims only adapting BN parameters, whereas the above reply seems to mention all the parameters.
> > 2. If MEME only adapts the BN parameters, I think it's better to integrate Section 3.2's "Adapting BN statistics." paragraph into Section 3.1 because it is an important MEME feature. Otherwise, readers will assume using all the model parameters when reading Section 3.1.
> > 3. I agree with Reviewer ZgMY on testing Tent with batch size 1 and model resetting. The key point is to redesign experiments to make different methods to use the same protocols. Of course, you can try different protocols, with each applied to all methods.

---

> ### Author Response · Authors · 2021-11-21
> **Followup response to reviewer 6Xgd**
>
> Thank you for your response! We are glad that you feel that your original comments have been addressed. We are following up with an additional experiment which investigates the efficiency of MEME, as requested, and we wish to respond to your remaining comments.
>
> > “Any investigation on efficiency? It would be better to add some discussions on how to improve efficiency.”
>
> We have added in Appendix B.2 (Figure 3) an experiment on ImageNet-R in which we vary the number of augmentations B from [1, 2, 4, 8, …, 128] and plot seconds per evaluation (x axis) against test error (y axis). We notice that significant accuracy gains are obtained for as few as 4 augmentations, indicating that a practical tradeoff between efficiency and accuracy is possible. For large B, the wall clock time is dominated by computing the augmentations -- in our implementation, we do not compute augmentations in parallel, though in principle this is possible for AugMix and should improve efficiency overall. We have also expanded our existing discussion of efficiency in Section 5, which highlights selective adaptation as a interesting direction for future work.
>
> > “I am still a little confused about what parameters MEME adapts.”
>
> We apologize that our revision was still unclear in this regard. We have re-revised Section 3.2 to simply use our language above: “we adapt the BN statistics alongside all of the model parameters for our method”. That is, single point BN adaptation updates \mu and \sigma^2, and MEME adapts all of \theta. For comparison, Tent also updates \mu and \sigma^2 (though with a batch of test points) but only adapts \gamma and \beta, which are the affine parameters in the BN layers, and not the rest of \theta. We hope that this point is now clear, please let us know if not.
>
> > “I agree with Reviewer ZgMY on testing Tent with batch size 1 and model resetting.”
>
> We have added experiments in Appendix B.1 (Table 4) in which we evaluate Tent with a batch size of 1 with model resetting. Tent typically uses BN adaptation on the batch of test points to update \mu and \sigma^2. Analogously to MEME, as described in Section 3.2, here we use a prior strength of N=16 for Tent’s BN adaptation, to prevent overfitting to the single test point. These results show that this method does not provide any additional performance gains over just using the single point BN adaptation. This shows that a naïve application of Tent is not a suitable approach for the test time robustness setting. Table 4 is too large to reproduce here, but as an example, here are the baseline ResNet-50 results:
>
> |  | ImageNet-C mCE | ImageNet-R Error (%) | ImageNet-A Error (%) |
> | :--- | ---: | ---: | ---: |
> | Baseline ResNet-50 | 76.7 | 63.9 | 100.0 |
> | Single point BN | 71.4 | 61.1 | 99.4 |
> | MEME (ours) | **69.9** | **58.8** | **99.1** |
> | Tent (batch size 1, with model resetting) | 71.4 | 61.2 | 99.4 |
>
> We believe that this response should address all of your points, but please let us know if you have any additional questions or concerns.

---

### Official Review · Reviewer_fKFu · 2021-11-03

**Correctness:** 1
**Technical Novelty And Significance:** 2
**Empirical Novelty And Significance:** 3
**Recommendation:** 6
**Confidence:** 4

**Main Review:**

Strengths:
* Motivation is clear
* Well-written and structured
* Addressing an important and challenging problem setting
* No assumptions on the availability of the entire test dataset
* A simple approach that is easy to implement
* Provided ablation study to understand the importance of the augmentations, confidence maximization and encouraging consistent predictions.
* Significant improvements are noticed on vision transformer model that are less reliant on BatchNorm layers, e.g., about 8% improvement on ImageNet-C and ImageNet-R over the baseline.

Weaknesses:
My major concerns lie on author's claims, novelty of the approach and effectiveness of the approach on CNNs.
* Novelty of the approach is limited. Prior works have utilized augmentations during test time and entropy minimization for test time adaptation independently. This work can be seen as combination of these two existing approaches for test time adaptation.
* Results on CIFAR-10 are similar or comparable to TTA.
* Individual analysis of corruptions and severities on ImageNet-C reveal that improvements are marginal and range between 0%-2%.
* Authors emphasize that invariances across the augmentation is significant but do not reflect that in the results, e.g., results in Table 3 show that major gains are obtained from confidence maximization of multiple augmentations but encouraging the invariance produce only marginal improvements. I assume similar behavior on ImageNet-C: considerable improvements are seen with confidence maximization of multiple augmentations but only marginal improvements by enforcing invariance.
* TENT produce worse results on three ImageNet benchmarks with vision transformer. Since this architecture is less reliant on BatchNorm layers, how do updating all the parameters of this model would perform with TENT?


**Summary Of The Paper:**

The paper focus on single image test time adaptation of deep neural network for distribution shift in classification task. Prior works perform test time adaptation on a batch of images or entire test dataset to capture the distribution statistics. The authors propose to create augmented copies of a provided test input and encourage invariance across different augmentations by promoting same prediction. Authors propose to minimize marginal entropy for confident predictions across the augmented versions of the input. Improvements are shown on CIFAR-10-C, CIFAR-10.1, ImageNet-C, ImageNet-R and ImageNet-A.


**Summary Of The Review:**

Given the simplicity of the approach for single image test time adaptation, its improvements on vision transformer and my concerns listed above, I marginally accept this paper. I am willing to reconsider my rating based on other reviews and authors responses.

-------- post rebuttal ---------------

The paper studies an interesting problem setting and also bring improvements in this setting, particularly on vision transformer. However, my concerns on authors claims still persist after the rebuttal and I also see that the novelty of the paper is limited. After considering the authors response and other reviewers comments, I rate this paper as borderline and keep my original score.

---

> ### Author Response · Authors · 2021-11-21
> **Response to reviewer fKFu**
>
> Thank you for your insightful comments. We have run the following requested experiments: additional ablations for confidence maximization on ImageNet and adapting all model parameters for the vision transformer and Tent. We have added these results to the paper and summarize them below. We also wish to respond to your other concerns regarding novelty and relative improvements of MEME.
>
> > “This work can be seen as combination of these two existing approaches for test time adaptation.”
> > “I assume similar behavior [of the confidence maximization ablation] on ImageNet-C: considerable improvements are seen with confidence maximization of multiple augmentations but only marginal improvements by enforcing invariance.”
>
> As requested, we have updated Table 3 with additional results for the confidence maximization ablation (which corresponds to the simple “combination of two existing approaches”) on the ImageNet test sets, focusing on the robust vision transformer model. These results show that confidence maximization also helps for the ImageNet problems (e.g., -8.2 mCE over the baseline on ImageNet-C), but MEME still performs better on ImageNet-C (-8.8 mCE) and ImageNet-R. The consistency with which MEME outperforms simple confidence maximization is significant and demonstrates that additionally encouraging invariance to augmentations leads to a more successful algorithm.
>
> |  | ImageNet-C mCE | ImageNet-R Error (%) | ImageNet-A Error (%) |
> | :--- | ---: | ---: | ---: |
> | RVT*-small | 49.4 | 52.3 | 73.9 |
> | MEME (ours) | **40.6** | **43.8** | 69.8 |
> | Confidence maximization | 41.2 | 44.2 | **69.7** |
>
> > “Results on CIFAR-10 are similar or comparable to TTA. Individual analysis of corruptions and severities on ImageNet-C reveal that improvements are marginal and range between 0%-2%.”
>
> We agree that most improvements over the best baseline are small. We believe that the main benefit of MEME is that it is *consistent* in providing these improvements, unlike the prior methods. In other words, the best baseline changes across experiments: TTA works well for ImageNet-A but actually *decreases* performance on ImageNet-C, and single point BN is good for ImageNet-C but not ImageNet-A. We have updated Section 4 to better highlight what we view as a major advantage of MEME.
>
> > “how do updating all the parameters of [the robust vision transformer] would perform with TENT?”
>
> Thank you for this suggestion! We have run this version of Tent and updated Table 2 with the results. We report results from both versions of Tent because we found that, compared to standard Tent adaptation, adapting all parameters improves results for ImageNet-C (-2.1 relative mCE improvement) and ImageNet-A (-1% relative error) but is catastrophic for ImageNet-R (+23.4% relative error). We noticed that this drop was largely due to entropy minimization changing the model too aggressively later on in the evaluation loop, and we believe that this indicates the careful design decisions that must be made when performing continual adaptation. Comparatively, MEME does not require these design decisions as we consider the test time robustness (single test point) setting, and for the simple choice of adapting all parameters, MEME is successful for all three ImageNet datasets.
>
> |  | ImageNet-C mCE | ImageNet-R Error (%) | ImageNet-A Error (%) |
> | :--- | ---: | ---: | ---: |
> | RVT*-small | 49.4 | 52.3 | 73.9 |
> | MEME (ours) | **40.6** | **43.8** | **69.8** |
> | Tent (standard) | 46.8 | 50.7 | 82.1 |
> | Tent (adapt all) | 44.7 | 74.1 | 81.1 |
>
> Thank you for helping to improve the paper! We believe that this response should address all of your points, but please let us know if you have any additional questions or concerns.

---

> > ### Comment · Reviewer_fKFu · 2021-12-06
> > **Response to authors comments**
> >
> > I thank the authors for providing additional results. These results confirm that major gains are obtained from confidence maximization of multiple augmentations and consistent but marginal improvements are seen by encouraging the invariance across the augmentations. It's interesting to study the reasons for the improvements through multiple augmentations, like do these multiple augmentations provide better statistics to stabilize bachnorm than a single input? Authors promote invariance across augmentations as one of their primary contribution, however the results weakly support their claim. Hence I stick to my initial assessment.

---

### Decision · Program_Chairs · 2022-01-20

**Decision:**

Reject

**Comment:**

The reviewers agree that test-time model adaptation is an interesting problem, providing a new perspective to improve model robustness. The proposed method builds on intuitive assumptions that are easy to understand. However, there are mainly two concerns regarding novelty and effectiveness. The paper can improve in these two aspects to meet ICLR standard.